

# Tree-based mesh-refinement GPU accelerated tsunami simulator for real time operation

Marlon Arce Acuña[1], Takayuki Aoki[2]

[1]Department of Nuclear Engineering, Tokyo Institute of Technology, 2-12-1-i7-3, Ookayama, Meguro, Tokyo, Japan
[2]Global Scientific Information and Computing Center, Tokyo Institute of Technology, 2-12-1-i7-3, Ookayama, Meguro, Tokyo, Japan

*Correspondence to*: Marlon Arce Acuña (marlon.arce@sim.gsic.titech.ac.jp)

**Abstract.** This paper presents a fast and accurate tsunami real time operational model to compute across-ocean wide-simulations completely on GPU. The spherical shallow water equations are solved using the method of characteristics
and upwind cubic-interpolation, to provide high accuracy and stability. A customized, user interactive, tree based mesh refinement method is implemented based on distance from the coast and focal areas to generate a memory efficient domain with resolutions of up to 50m. Three GPU kernels, specialized and optimized (wet, wall and inundation) are developed to compute the domain block mesh. Multi-GPU is used to further speed up the computation and a weighted Hilbert space filling curve is used to produce balanced work load. Hindcasting of the 2004 Indonesia tsunami is presented to validate and
compare the agreement of the arrival times and main peaks at several gauges. Inundation maps are also produced for Kamala and Hambantota to validate the accuracy of our model. Test runs on three Tesla P100 cards on Tsubame 3.0 could fully simulate 10 hours in just under 10 minutes wall clock time.

## 1 Introduction

The turn of the 21st century showed us as never before the reality of the terrible and devastating damage and death that
tsunamis can cause. In 2004, a massive earthquake of magnitude M9.0 on the Richter scale off Sumatra Island triggered a tsunami with deadly consequences. According to the World Health Organization the death toll for these events exceeds 200,000 (WHO, 2014) in several countries spread along the Indian Ocean. Not much later in 2011 a tsunami triggered by a M9.0 earthquake on the east coast of Japan produced in the Tohoku region yet another disaster. Over 15,000 people died from these events with massive destruction in port and city infrastructure, housing, tele-communications and the subsequent
nuclear crisis due to the tsunami-induced damage of several reactors in the Fukushima nuclear power plant (Motoki and Toshihiro, 2012).

These events highlight the importance of developing accurate and fast tsunami forecasting models. For several decades, efforts have been made to develop such models. These can be classified in two main groups: depth-average, hydrostatic and non-hydrostatic long wave equations. Hydrostatic models for the shallow water equations (SWE) started by solving their
linear form based on finite difference methods (FDM) taking after the work of (Hansen, 1956) and (Fischer, 1959) in the



1950s. The model TUNAMI (Tohoku University's Numerical Analysis Model for Investigation) (Imamura et al., 1995) came from these initial steps but solved the shallow water equations in a non-linear form instead, formulated in a flux-conservative way for mass conservation and also introduced a discharge computation (Imamura, 1996) for the elevation near the shoreline. In a very similar manner the ALASKA-tectonic and Landslide models (GI'-T) (GI'-L) were introduced, also

solving the non-linear shallow water and using leapfrog FDM (Nicolsky et al., 2011) as TUNAMI. Later came MOST (Method of Splitting Tsunami) (Titov and Synolakis, 1995), an extensively used model for tsunami simulation, that tried to incorporate the effect of dispersion during simulation (Burwell et al., 2007), also it was original by introducing a function to add points in the shoreline to keep better tracking. A more recent model is GeoClaw which implements a unique approach to deal with the issue of transferring fluid kinematic throughout nested grids by refining specified cells during simulation

getting better resolution in those areas (Berger and LeVeque, 1998). More recent models incorporate a real-time application such as RIFT (Real-Time Inundation Forecasting of Tsunamis) (Wang et al., 2012). Like several of the previous models a leap-frog scheme is also used for these real-time models and a linear SWE is solved in certain areas for lighter computation. EasyWave is another known model (Babeyko, 2017), which employs linear approximations for speed up and leap-frog scheme as its numerical scheme. The latest version of EasyWave introduced GPU to accelerate parts of the existing CPU

code.

       In order to include the effect of pressure, since the 1990s some models took the direction of solving non-hydrostatic models using the depth-integrated Boussinesq equations (BE) instead of the SWE for tsunami propagation. Initial efforts considered a weak nonlinear model (Peregrine, 1967) however, models for the nonlinear equations were also developed not long after, for instance (Nwogu, 1993), (Lynett et al., 2002). Solving the Boussinesq equation is in general more

computationally demanding than solving the SWE and in order to reduce the computational time some techniques have been implemented, such as using parallel clusters or introducing nested-grids. An example of this is FUNWAVE-TVD (Shi et al., 2012), which is an extended version of FUNWAVE, a run-up and propagation model based on fully nonlinear and dispersive Boussinesq equations (Wei et al., 1995). FUNWAVE introduced a nested grid method and its later version has been fully parallelized using MPI-FORTRAN. Another of these models is BOSZ (Roeber and Cheung, 2012) ), which combines the

dispersive effect from the BE with the shock capturing ability of the nonlinear SWE. BOSZ is mainly used for near-shore simulation since is based on Cartesian coordinates and not suited for large areas, also it does not implement nested grids. Recently, efforts to solve the modelling equations in three dimensions have been made as well. Although these models tend to capture difficult coastlines very well and can include multiple fluids or even materials, the computation cost is still so great that it makes it possible to apply them effectively only in small areas and not viable for transoceanic propagations.

Some examples are SELFE (Semi-Implicit Eulerian-Lagrangian Finite Elements) (Zhang and Baptista, 2008), (Abadie et al., 2010), (Horrillo et al., 2013) and (Abadie et al., 2012).

       In this work we present a new approach for a tsunami operational model that retains a high degree of the complexities of the physics involved, and delivers a fast and accurate simulation. This speed also enables real-time operation: a user can start forecasting simultaneously as a tsunami event occurs. Results are generated faster-than-real-time. The main goal is to



accomplish a wide-area, ocean-size, computation in short time and using resources efficiently. Our model, referred to hereinafter as *TRITON-G (Tsunami Refinement and Inundation Real-Time Operational Numerical Model for GPU)*, implements a full-GPU computing approach for the whole tsunami model, composed of generation-propagation-inundation. Specialized kernels are developed for each part of the tsunami computation and multi-GPU is used for further acceleration.

Load balance is obtained using a weighted Hilbert space filling curve. TRITON-G solves the non-linear spherical shallow water equations across the entire domain to preserve the complexity of the propagation and effects near the coastline. The method of characteristics with directional splitting and a 3rd Order Interpolation Semi-Lagrangian numerical scheme is used to solve the governing equations. This allows high accuracy and minimizes effects of numerical dispersion and diffusion, also give the ability of choosing a larger time step compared to that of using a Runge-Kutta scheme and at the same time

permits a light stencil suitable for fast computation. We implement a tree-based block refinement to generate a computational mesh that is flexible, light and can track complex coastlines. Customized refinements by distance and focal area were developed, which permits an efficient use of memory and computational resources. In a collaboration project with RIMES (Regional Integrated Multi-hazard Early Warning System) (RIMES, 2017) we utilize their existing databases for bathymetry and fault sources where available, and successfully deployed TRITON-G as their tsunami forecast operational

model.

This article is organized as follows, a review of the governing equations is given in Section 2. The numerical method and boundaries are explained in Section 3. In Section 4 a description of tree-based refinement and its customization is given. Topography and bathymetry used are also described. GPU and parallel computing is covered in Section 5. In Section 6 we present several numerical results including TRITON-G validation with existing tsunami propagation data and run-up

measurements. Section 7 presents the conclusions of this study.

## 2 Governing Equations

The spherical non-linear shallow water equations (SSWE) are used to compute the tsunami propagation. In specific and small areas where inundation needs to be computed, the Cartesian coordinate version of the SWE are solved instead, see (Toro, 2010). The SSWE (Williamson et al., 1992), (Swarztrauber et al., 1997) can be written as

$$\frac{\partial h}{\partial t} + \frac{1}{a\cos\theta}\frac{\partial}{\partial\lambda}(hu) + \frac{1}{a}\frac{\partial}{\partial\theta}(hv) - \frac{hv}{a}\tan\theta = 0,$$

$$\frac{\partial hu}{\partial t} + \frac{1}{a\cos\theta}\frac{\partial}{\partial\lambda}\left(hu^2 + \frac{g}{2}h^2\right) + \frac{1}{a}\frac{\partial huv}{\partial\theta} - \frac{hv}{a}\tan\theta - \left(f + \frac{u}{a}\tan\theta\right)hv + \frac{gh}{a\cos\theta}\frac{\partial z}{\partial\lambda} + \tau_\lambda = 0, \tag{1}$$

$$\frac{\partial hv}{\partial t} + \frac{1}{a\cos\theta}\frac{\partial hvu}{\partial\lambda} + \frac{1}{a}\frac{\partial}{\partial\theta}\left(hv^2 + \frac{g}{2}h^2\right) - \frac{hv^2}{a}\tan\theta + \left(f + \frac{u}{a}\tan\theta\right)hu + \frac{gh}{a}\frac{\partial z}{\partial\theta} + \tau_\theta = 0$$





where λ stands for the longitude coordinate, θ for the latitude coordinate, $h$ is the water depth, $hu$ and $hv$ are the momentum in longitude and latitude respectively with corresponding velocities u and v, g is gravity, a is the radius of the Earth, z is the bathymetry (submarine topography), $f$ is the Coriolis force defined as $f = 2\Omega \sin\theta$ with Ω being the rotation rate of the Earth and $\tau$ is the bottom friction term. The bottom friction is determined using the Manning formula

$$\tau_\lambda = \frac{gn^2}{h^{7/3}} hu\sqrt{(hu)^2 + (hv)^2},$$

$$\tag{2}$$

$$\tau_\theta = \frac{gn^2}{h^{7/3}} hv\sqrt{(hu)^2 + (hv)^2}$$

where $n$ is the Manning's roughness coefficient, the default value used for $n$ is 0.025 across all domain except for specific areas where more detailed values in the coastline are given in a database. The parameters used in this work are $a = 6.37122 \times 10^6 \, [m]$, $\Omega = 7.292 \times 10^{-5} \, [s^{-1}]$ and $g = 9.81 \, [ms^{-2}]$.

## 3 Numerical methods and boundary conditions

### 3.1 Methods of characteristics for SSWE

The SSWE are solved using the method of characteristics (MOC). A method developed in the 1960s, explained in detail by Rusanov (Rusanov, 1963). MOC is applied to reduce hyperbolic partial differential equations, such as the SSWE, to a family of ordinary differential equations. A traditional approach when using MOC is to introduce a dimensional splitting (Nakamura et al., 2001) in the 2-dimensional equations to create a smaller stencil and lighter computation. A numerical scheme is regarded as well-balanced, or satisfying the C-property (Bermúdez and Vázquez, 1994) if it preserves steady states at rest, for instance, the undisturbed surface of lake. When the fluid is at rest i.e. $u(x,t) = 0$ then the constant water height $H$ defined as $H(x,t) = h(x,t) + z(x)$ represent a steady state that should hold in time and not produce spurious oscillations (LeVeque, 1998). In order to make the model well-balanced, the SSWE equations are solved for $H$ during the simulation to guarantee this steady state. The original variable $h$ is simply obtained back from the expression $h = H - z$.

In order to apply the method of characteristics, first the SSWE Eq. (1) are re-written in vector form as




$$\frac{\partial \boldsymbol{U}}{\partial t} + \boldsymbol{A}\frac{\partial \boldsymbol{U}}{\partial \lambda} + \boldsymbol{B}\frac{\partial \boldsymbol{U}}{\partial \theta} + \boldsymbol{S} = 0 \tag{3}$$

with

$$\mathbf{U} = \begin{bmatrix} h \\ hu \\ hv \end{bmatrix}$$

$$\mathbf{A} = \frac{1}{a\cos\theta}\begin{bmatrix} 0 & 1 & 0 \\ \Gamma^2 - u^2 & 2u & 0 \\ -uv & v & u \end{bmatrix}$$

$$\mathbf{B} = \frac{1}{a}\begin{bmatrix} 0 & 0 & 1 \\ -uv & v & u \\ \Gamma^2 - v^2 & 0 & 2v \end{bmatrix}$$

$$\mathbf{S} = \begin{bmatrix} \dfrac{-hv\tan\theta}{a} \\ -\left(f + \dfrac{u}{a}\tan\theta\right)hv - \dfrac{huv}{a}\tan\theta + \dfrac{gh}{a\cos\theta}\dfrac{\partial z}{\partial \lambda} \\ \left(f + \dfrac{u}{a}\tan\theta\right)hu - \dfrac{hv^2}{a}\tan\theta + \dfrac{gh}{a}\dfrac{\partial z}{\partial \theta} \end{bmatrix}$$

where $\Gamma \equiv \sqrt{gh}$. Using the directional splitting technique on Eq. (1) three equations are produced. An equation for each coordinate, longitude $\lambda$, and latitude $\theta$, and the third for the source term $S$. The latter equation simply represents an ordinary partial differential equation for the source term while Eq. (4) and Eq. (10) for the coordinates are in advection form. These last two equations are written in diagonal form in order to find the Riemann invariants and characteristics curves, a detailed description of this procedure can be found in (Ogata and Takashi, 2004) or (Stoker, 1992). The equation for the longitude coordinate $\lambda$ given by

$$\frac{\partial \mathbf{U}}{\partial t} + \boldsymbol{A}\frac{\partial \mathbf{U}}{\partial \lambda} = 0 \tag{4}$$





has eigenvalues $\Lambda$ given by

$$\Lambda_{\pm}{}^{\lambda} = \frac{1}{a\cos\theta}(u + \Gamma), \quad \Lambda_3{}^{\lambda} = \frac{1}{a\cos\theta}u, \tag{5}$$

5   which inserted in the diagonal form of Eq. (4) leads to

$$\frac{D^{\pm}}{Dt}(\Gamma \pm \frac{u}{2}) = 0 \tag{6}$$

where D/Dt represents the material derivative. Equation (6) means that the solution at a given grid point *i* is determined from two characteristic curves along $C^+$ and $C^-$ (Fig. 1). The result at a time *n+1* can be found by adding and subtracting the

10   expressions in Eq. (6) respectively to obtain

$$\Gamma_i{}^{n+1} = \frac{1}{2}\left\{\Gamma^+ + \Gamma^- + \frac{1}{2}(u^+ - u^-)\right\}, \tag{7}$$

$$u_i{}^{n+1} = \frac{1}{2}\{u^+ + u^- + 2(\Gamma^+ + \Gamma^-)\} \tag{8}$$

where $\Gamma^{\pm}$ and $u^{\pm}$ are the values at a time *n*, at positions which might not necessarily lie on a grid point. An interpolation in applied in order to determine these values and with them solve Eq. (7) and Eq. (8).

Following a similar procedure as (Yabe and Aoki, 1991), (Yabe et al., 2001), (Utsumi et al., 1997) we utilize a cubic-polynomial approximation on the grid profile to find the interpolated values. The polynomial is defined as

$$F(\lambda) = a\lambda^3 + b\lambda^2 + c\lambda + d \tag{9}$$

with





$$u\Delta t > 0 \begin{cases} a = \dfrac{f_{i+1} - 3f_i + 3f_{i-1} - f_{i-2}}{6\Delta\lambda^3} \\[2ex] b = \dfrac{f_{i+1} - 2f_i + f_{i-1}}{2\Delta\lambda^2} \\[2ex] c = \dfrac{2f_{i+1} + 3f_i - 6f_{i-1} + f_{i-2}}{6\Delta\lambda} \\[2ex] d = f_i \end{cases}$$

$$u\Delta t \leq 0 \begin{cases} a = \dfrac{f_{i+2} - 3f_{i+1} + 3f_i - f_{i-1}}{6\Delta\lambda^3} \\[2ex] b = \dfrac{f_{i+1} - 2f_i + f_{i-1}}{2\Delta\lambda^2} \\[2ex] c = \dfrac{-f_{i+2} + 6f_{i+1} - 3f_i - 2f_{i-1}}{6\Delta\lambda} \\[2ex] d = f_i \end{cases}$$

5    A similar analysis can be made for the latitude equation $\theta$ obtained from the splitting method, given by

$$\frac{\partial \mathbf{U}}{\partial t} + B \frac{\partial \mathbf{U}}{\partial \theta} = 0 \tag{10}$$

with analogous results for the eigenvalues and curves

$$\Lambda^\theta{}_\pm = \frac{1}{a}(v + \Gamma), \quad \Lambda^\theta{}_3 = \frac{1}{a}v, \tag{11}$$

$$\frac{D^\pm}{Dt}\left(\Gamma \pm \frac{v}{2}\right) = 0 \tag{12}$$

From which similar expressions as Eq. (7) and Eq. (8) can be found in order to estimate the values for $h$ and $hv$.

The equations for the coordinates are solved using the fractional step method. Following this method, the source term given by

$$\frac{\partial \mathbf{U}}{\partial t} + S = 0 \tag{13}$$




is added to the solution obtained for Eq. (4) and Eq. (10). For the source term, central finite differences are used to solve the bathymetry term while the remaining values (cosine, tangent terms) can be solved analytically at each grid point since the variables are known straightforwardly.

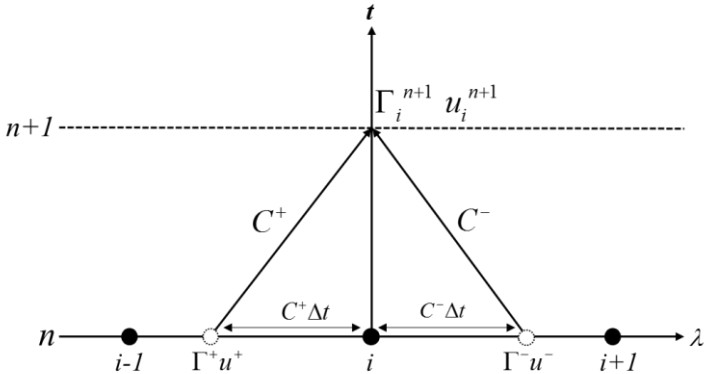

**Fig. 1 Space-time diagram showing the characteristic curves $C^{\pm}$ where black dots represent the grid points, dotted points represent the values $\Gamma^{\pm}$ and $u^{\pm}$ to be interpolated to find $\Gamma^{n+1}$ and $u^{n+1}$.**

In order to validate the implementation of the numerical methods for the SSWE, we used the benchmark described in (Kirby et al., 2013), where an initial Gaussian wave is propagated on an idealized sphere with water depth $h$=3000m. Results after 5000s show good agreement with the results reported which confirms the accurate propagation of the wave on the sphere and the effects of the curvature and Coriolis force.

**3.2 Run-up calculation**

The Cartesian SWE are solved in specific areas of just a few kilometers where inundation has to be calculated. For this case we use a finite volume implementation (Bradford, 2002), (LeVeque and George, 2014) briefly described here. The Surface Gradient method (SGM) (Zhou et al., 2001) is utilized to solve the SWE. This method uses the data at cell center to determine the fluxes. In general, depth gradient methods cannot accurately determine the water depth value at cell interface, since effects of the bed slope or small variations in the free surface cannot be determined accurately. These inaccuracies are spread during the computation resulting in an incorrect simulation of the inundation. In order to overcome this, the SGM uses a constant water level $H$. Figure 2 depicts the stencil for the water depth reconstruction, by using the constant $H$ as the total water depth at the cell interface ($i$+0.5) instead, the water depth be can determined accurately. In order to reconstruct the water depth the following expression is used

$$h_{L,R\ i+0.5} = \max(H_{L,R\ i+0.5} - \overline{z}_{i+0.5}, 0) \tag{14}$$



where $\bar{z}$ is given by

$$\bar{z}_{i+0.5} = (z_i + z_{i+1})/2 \qquad (15)$$

A MUSCL scheme (Yamamoto and Daiguji, 1993) is used to find the flux value while Local-Lax-Friedrichs (LeVeque,
5   2002) is used to solve the bed slope source term. For the time integration a 3rd Order TVD Runge-Kutta scheme was used.
Lastly, the bottom friction is computed using Manning's formula.

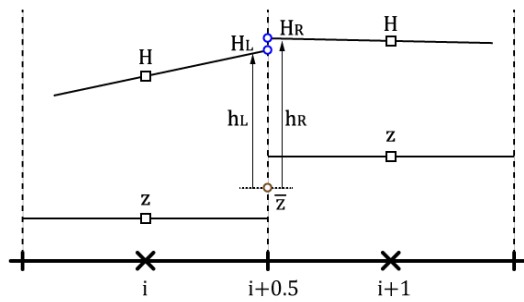

Fig. 2 Reconstructed water depth $h_{L,R}$ for inundation (LeVeque and George, 2014).

This run-up implementation assumes a thin film of water on land defined as ε. This parameter, set much smaller
compared to the wave height, allows the computation of the wave inundation over land while keeping it stable. If the water
height is less than ε (i.e. $h<\varepsilon$) then the height value is fixed as ε and the momentum is set as rest (i.e. *hu=hv=0*) on that grid
point. This implementation has proven to be robust and stable under different benchmarks and simulations.

15   The one-dimensional dam break benchmark (Stoker, 1992) was used to compare the results with its analytical solution
and good agreement was found. The shock wave was successfully captured for different initial water heights. The Parabolic
Bowl problem proposed by (Thacker, 1981) was also used to compare the accuracy of the inundation. The bottom
bathymetry is given by

$$z(r) = -D_0(1 - \frac{r^2}{L^2}) \qquad (16)$$

while the water height at a time *t* can be found from the analytical solution





$$H(r, t) = D_0 \left( \frac{(1 - A^2)^{\frac{1}{2}}}{1 - A \cos \omega t} - 1 - \frac{r^2}{L^2} \frac{1 - A^2}{(1 - A \cos \omega t)} \right),$$

$$r = (x - L_x/2)^2 + \left(y - L_y/2\right)^2,$$

$$\omega = \sqrt{8gD_0/L^2},$$  (17)

$$A = \frac{(D_0 + \eta)^2 - D_0{}^2}{(D_0 + \eta)^2 + D_0{}^2}.$$

We use these parameters $L_x=L_y=8000$, $L=2500$, $D_0 = 1$ and $\eta=0.5$. Two grid sizes were used for testing, 80×80 and 160×160. Figure 3 shows the oscillating water in the bowl at different times. As it can be seen, the inundation method is able to capture well the analytical solution of the water height as it evolves in time on the different grid sizes. Measurements on

5    this tests showed a third-order reduction of the error as the value ε was decreased.

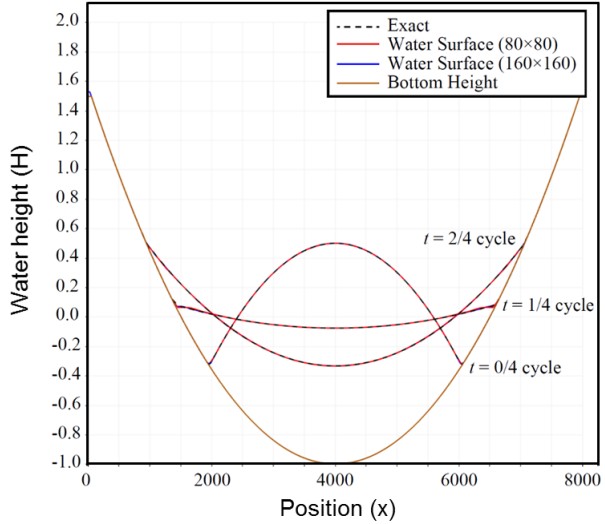

**Fig. 3 Parabolic bowl problem cross section with $\epsilon = 10^{-4}$.**



### 3.3 Tsunami source model

TRITON-G follows the three-step simulation model: *Generation-Propagation-Inundation*. In the *generation* process, a good initial source model is essential in order to obtain an accurate simulation. However due to the complex nature of the source

dynamics during an earthquake and the difficulty to track it in real time (as it happens), currently it is beyond our grasp to obtain these parameters precisely and instantly. For these reasons we opted for a coseismic deformation. This deformation is calculated from the theory of displacement fields proposed by (Smylie and Mansinha, 1971) . Their objective is to provide a closed analytical expression that "facilitates the interpretation of near-fault measurements". The expressions provided, valid at depth and surface, consist solely on algebraic and trigonometric functions that can be readily evaluated numerically based

on a few source parameters like dip, strike, slip and length. These values are obtained from RIMES' databases online or loaded from a file. The original source generation code, provided by RIMES, was written for CPU and ported by us to GPU for this study.

### 3.4 Boundary conditions

Two kind of boundary conditions are used, *open and closed*. Open boundary sets conditions to allow waves from within the model to leave the domain through an edge without affecting the interior solution. *Closed* boundary which keep the fluid inbound in the domain, physically it means that no water flows across the edges. A wall boundary condition creates a physical total reflection when a wave hits a dry point.

In Eq. (1) the term $\cos\theta$ in their denominators produces a singularity at the poles of the spherical coordinate system. When working on a complete sphere, special techniques and treatment are required to compute values over the poles without divergence. In this study, the domain chosen represents a portion of the Earth centered in the Indian Ocean and does not extend near the poles in any circumstance which permits us to avoid this pole singularity.

The boundaries for the computational domain are set as open boundary condition at the South and East edges, and closed

boundary condition at the North and West edges. All coastlines have wall boundary condition except for the special cases where particular regions set as *inundation* are defined. In those cases a complete run-up is computed using the methods described in previous sections. Since the inundation method is relatively computationally intensive, using two kinds of boundaries in the coasts permits to focus computational resources just in areas of interest.

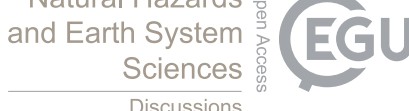



## 4 Tree-based mesh refinement and bathymetry

An efficient use of resources, memory and computation, requires a mesh that covers areas of interest with high resolution
only where desired, and leaves the rest of the domain coarser. The concept of this approach is similar to that of the adaptive
5    mesh refinement, initially introduced by (Berger and Oliger, 1984), (Colella, 1989) in the 1980s as a method to solve PDEs
on an automatically changing hierarchal grid, solving for a set accuracy on certain areas of the interest instead of
unnecessarily overly refining on the entire domain.

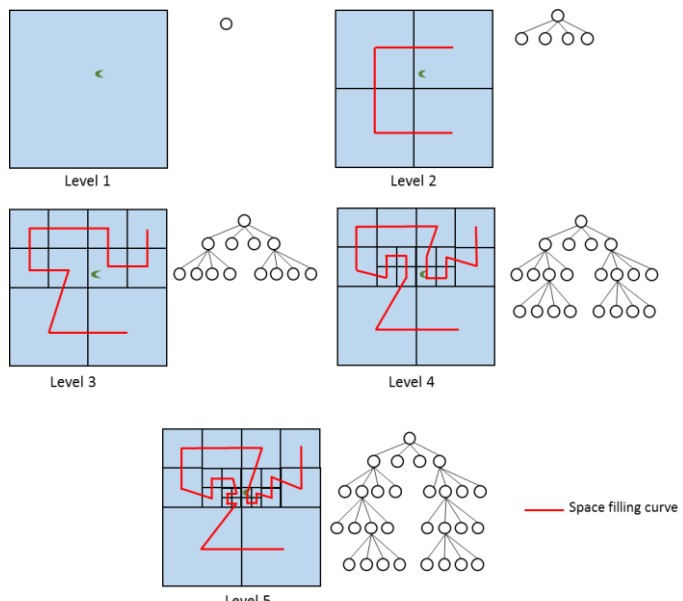

**Fig. 4 Tree-Based block refinement with quadtree structure and Hilbert space filling curve for 5 levels**

To generate the mesh for the domain, we use a customized tree-based mesh refinement, without the need of re-meshing
during simulation since the geometrical features remain unchanged. We briefly explain the process of tree-based refinement
(Yerry and Shephard, 1991). Figure 4 illustrates this procedure using a moon-shaped green point as the area of interest. At
15   each level, the domain and its tree structure, called *quadtree*, is presented. Initially just a quadrant and its *quadtree* root exist.
Each quadrant represents a block of domain points. At level 2, one refinement has occurred and the original quadrant (father)
is replaced by four new ones (children). By containing the same number of points as their parent quadrant, these *children*
allow for greater resolution. Each child is represented as a *leaf* of the tree's root. Level 3 shows the refinement of two of the
Level 2 quadrants and are represented as two new leaves deeper on the *quadtree*. Focusing around the point of interest,
20   Levels 4 and 5 show the subsequent refinement of two quadrants of their respective previous levels. As it can be seen, each





refined quadrant is replaced by four new ones and these extend deeper on the tree. This process can continue recursively until reaching a desired goal, usually based on resolution or minimum error. Using this block refinement allows for greater resolution only around the points of interest while the *quadtree* data structure associate with it keeps track of the blocks connectivity.

The difference on spatial resolution between two adjacent levels is called the refinement ratio. For nested grids, this ratio is any positive integer. However using large integers tend to introduce inaccuracies in the computation. The existence of an abrupt change from one level to the next requires special boundary treatment, especially when complex bathymetry or topography is involved. For tree-based refinement this ratio is fixed as $\Delta x_l / \Delta x_{l+1} = 2$, where $l$ represents the block level and $\Delta x$ the grid resolution. This constant and small ratio creates a smooth wave transition between levels.

## 4.1 Customized mesh generation

The domain used for this work represents a large portion of the Indian Ocean (Fig. 5), which consists initially of a uniform mesh of $56 \times 30$ blocks, each made up of $65 \times 65$ node-centered cells. Using the tree-based refinement, specialized customizations are developed to adapt it to our specific needs. In general, mesh refinement methods utilize an error

estimation as the rule to determine if a block should be refined, however in this implementation the refinement depends on a *target* grid resolution combined with two factors, the block's distance from the coastline and the presence of a focal area.

The refinement rule's first factor depends on the distance of the block to the shoreline, the objective is to recursively refine blocks close to the coast until reach a target high-resolution threshold, while blocks far in the ocean remain with a coarser

resolution. This process involves two steps, determining the block's distance from the coast and checking if its distance is within refinement.

To accurately estimate the geo-distance between two points can be a complex task since the surface of the Earth is not a perfect sphere. However, for our refining purposes, a rough estimate is enough to determine the distances between the shoreline and the blocks. This is achieved by creating a signed distance function based on the Level-set method. A detailed

explanation of this procedure can be found in (Fedkiw and Osher, 2003). The distance function's *zero* level is represented by the cells along the shoreline ($z$=0). Positive distances represent cells on land while negative distances represent cells on the ocean. Using this distance values, each block is tested for refinement. Blocks with one cell or more within a certain distance from the coast, called *refinement stripe*, are flagged for refinement until they reach the fine-target resolution. The width of the refinement stripe is problem dependent and is input by the user based on their needs.



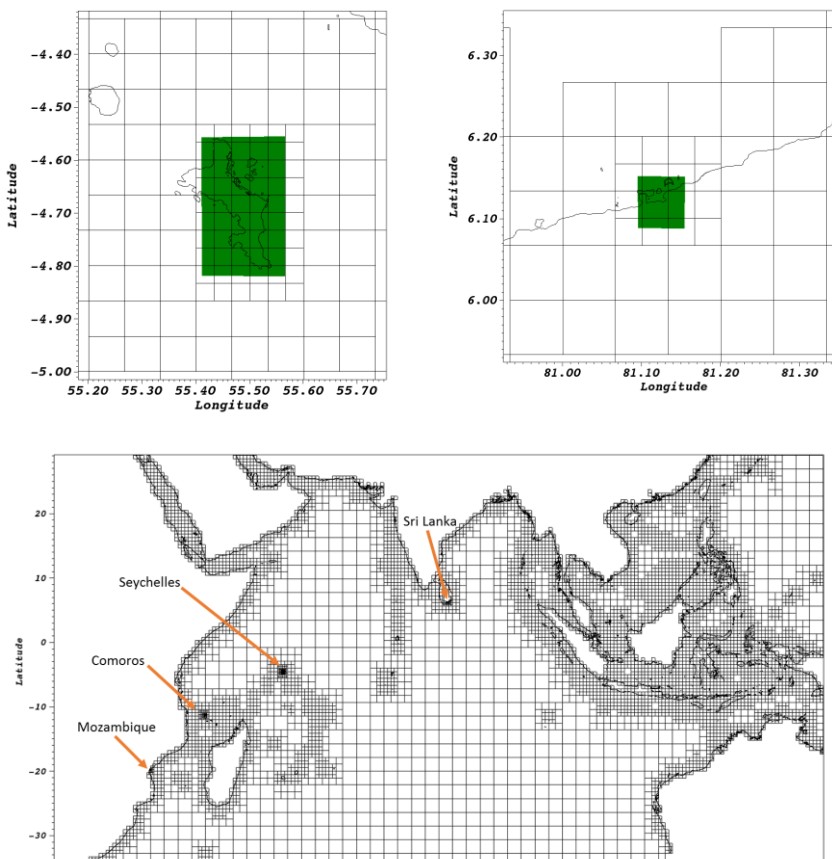

**Fig. 5 Bottom: Mesh Refinement for Indian Ocean Domain with 4 Focal Areas; Mozambique, Comoros, Seychelles and Sri Lanka. Top: Zoom on Sri Lanka and Seychelles regions, FA highlighted in green.**

For this study the initial resolution at ground Level 1 is 2 arc-min (an arc-minute being 1/60 of a degree, at Earth's equator equivalent to 1852 m) and the target finest resolution is 0.03125 arc-min (approximately 50 m), generating a total of 7 levels. This block refinement process can accurately trace complex coastlines and focus high resolution only in the shores. A downside is the considerably large number of total blocks generated, over 230,000 in initial tests, which represents over

10    100 GB of memory storage.

In order to reduce the memory footprint, we use the fact that only certain regions need high resolution, which inspired us to use a second refinement factor named *focal areas* (FA). This second factor is an additional constraint which consists in locating on the domain a convex polygonal area which serves as a refinement delimiter. It is possible to locate more than one at a time and since this is an additional constraint to the first refinement step, only blocks flagged for refinement at the first

15    step need to be tested again. On this second test, a block is tested if it is inside or outside a focal area. If a block is



completely outside the focal area, then it is un-flagged for refinement. Only blocks partially or totally inside the focal area are refined. The process of determining if a block lies inside or outside a focal area is based on collision detection theory using the Separating Axis Theorem (SAT). This is a well-known theorem applied to physical simulations (Szauer, 2017) and consists of a relatively *light* algorithm for 2D, which allows to test large number of blocks rapidly. A description of the SAT

can be consulted in (Moller et al., 1999) or (Gottschalk et al., 1996). Since the focal area is an additional constraint, it can be toggled active after any chosen level. A specific number of levels can be refined without this constraint while the following are affected and delimited. A last property of the FAs is that blocks with the highest resolution (Level 7) within, and that contain dry points represent inundation areas. This implies computing the complete run-up instead of using a closed boundary on the coastline.

The last step in the mesh generation consists in the removal of *land dry-blocks*. Considering that tsunami inundations, with few exceptions, generally extend tens to hundreds of meters inland, it becomes clear that blocks located deep inland are unnecessary for the computation. For this reason all blocks whose cells' distances are larger than a *land-distance* threshold are considered *land dry-block* and deleted from the domain.

The complete result of the customized refinement in the Indian Ocean domain is shown in Fig. 5. Four focal areas are

used located in Mozambique, Comoros, Seychelles and Sri Lanka. The focal area constraint start after Level 3. This value is chosen to coincide with GEBCO's 30 arc-second bathymetry, using the highest available accuracy for the coasts without needing to interpolate. The final result shows the refinement at higher levels limited to within the focal areas. All dry blocks exceeding the land-distance threshold of 10km were removed from the mesh. This reduced the number of blocks generated drastically to 7849, while the memory needed to store them became less than 15GB. This customized refinement procedure

proved to be fast and efficient, taking just around a minute to produce the results. The meshes generated by TRITON-G can be either computed real-time or loaded from a repository at the beginning of the simulation.

## 4.2 Halo exchange

Blocks must exchange results with their neighbors after each time step for the next iteration. For this purpose they share a boundary layer in their adjoining sides. This layer or *halo* extends over the neighbor's grid and updating represents one of three kinds of operation: copying, coarsening or interpolating.

If two neighbor blocks have the same level, then the halo is readily updated by exchanging values directly without any further computation, this represents a copying swap. If the neighbors are at different levels ($l$ and $l+1$) then additional

computation is required before the halo exchange. If the block's neighbor is one level up then values for the halo are averaged down from the block with higher accuracy before swapping, this has the effect of passing down better accuracy to blocks with lower resolution like in a cascade effect. The last case, interpolating, occurs when the block's neighbor is one



level down. For this, the values for the halo are interpolated from the neighbor block using a third-order polynomial interpolation, similarly as in Eq. (9). The portion of the boundary stencil used for interpolation is shown in Fig. 6.

(a)  West,

South cases

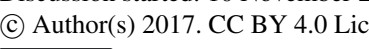

(b)  East,

North cases

**Fig. 6 Halo interpolation stencil for the four edges (a) west, south and (b) east, north.**

The new values for the halo for the north ($N$) and east ($E$) edges can be found from

$$f^{N,E}_{P1} = \frac{1}{4}(f_j + 4f_{j+1} - f_{j+2}),$$
$$f^{N,E}_{P2} = \frac{1}{4}(-f_j + 6f_{j+1} - f_{j+2})$$

(18)

15 since they are analogous orientations. For the south ($S$) and west ($W$) edges similar expressions are used

$$f^{S,W}_{P1} = \frac{1}{4}(-f_{j-2} + 4f_{j+1} + f_j),$$
$$f^{S,W}_{P2} = \frac{1}{4}(-f_j + 6f_{j+1} - f_{j+2})$$

(19)

In order to avoid spurious waves that might be generated from interpolating the water height value $h$, constant water level $H$ is used instead, and the original variable is recovered by using the relation $h = H - z$.

**4.3 Topography and bathymetry**

The data used in this study for bathymetry and topography comes from different sources. Initially, The General Bathymetric Chart of the Oceans (Oceans (GEBCO), 2017) database is used on the entire domain. GEBCO is freely available in 30 arc-
25 second spatial resolution. When coarser resolution is needed, values are averaged from this database. On the contrary, if finer





resolution is needed, a third order interpolation is implemented to generate the new values. Where available, databases with more precise measurements are used to replace the original GEBCO database. For the focal areas in Mozambique, Comoros, Seychelles and Sri Lanka, RIMES' proprietary databases generated from field measurements were provided to us to estimate the inundation more accurately.

## 5 GPU computing

The introduction of C-language extension CUDA (NVIDIA, 2017a) by NVIDIA® was a disruption in the traditional way simulations were done. By providing a way to program their graphic cards for scientific purposes (general purpose GPU, GPGPU), researchers no longer had to rely solely on CPU processors to perform calculations. Due to the intrinsic parallelism of graphics, GPUs naturally evolved to deliver in a card hundreds, and later, thousands of processors more than CPUs. The main reason behind the exceptional performance of GPUs lies in the specialized design for compute-intensive, highly parallel computation, with transistors dedicated exclusively to processing as opposed to flow control and data caching. The latest NVIDIA Tesla cards P100, with Pascal architecture have a peak performance of 9.3 Teraflops on single precision and 4.7 Teraflops on double precision (NVIDIA, 2017b). We take advantage of this technology to develop a full-GPU implementation to deliver fast forecasting results.

## 5.1 SSWE GPU kernels

A kernel is the style CUDA provides to define functions that get executed in parallel on GPU. Each kernel launch is organized in a grid of threads of CUDA blocks. The clear analogy between CUDA blocks and mesh blocks provided a guide to organize the grid for GPU execution. The SSWE are computed exclusively on GPU by processing the mesh blocks created during the domain refinement step and are stored in a structure of arrays on GPU global memory. Each mesh block have a size of $(65+4) \times (65+4)$, where the 4 corresponds to the total size of the halo. CUDA threads can be organized in any three-dimensional block configuration as needed by the problem. Since GPUs process threads in warps of 32, using multiples of this number is desirable to avoid performance penalties.

The kernel grid configuration for the SSWE is described briefly. For the $x$ dimension, CUDA threads are organized in two dimensional blocks of size: 64×4. Since the 64 threads in the $x$ dimension cover the length of a mesh block, only one CUDA block is needed. For the grid $y$ dimension, 16 CUDA blocks are requested for a total of 16×4=64 threads, covering the height of the mesh block and the combined CUDA $x$-$y$ blocks covering the area of 64×64. To process all the mesh blocks, this two-dimensional CUDA block configuration is extended along the $z$-direction as many times as mesh blocks exist. The computation of the $65^{th}$ cells is done separately with a specialized kernel based on the SSWE kernel.



In the case of Cartesian SWE kernel, the grid chosen for this kernel is different than that of the kernel for SSWE. In this case, a mesh block is sub-divided and covered by CUDA blocks of 16×16 threads. The excess of threads at the edges is not computed using a conditional limiting the grid size.

The source fault code was ported to GPU from the original C version. Due to the exclusively arithmetic operations and lack of a stencil memory access involved, a 20× speed up was achieved, reducing the computation of the initial condition to just a few seconds.

Several kernel optimizations were applied in order to accelerate the model's time-to-solution. This includes using the latest CUDA version to take advantage of the latest compiler updates. To avoid branch diversion as much as possible parts of the numerical method were re-written to eliminate conditionals. Precomputing terms that do not change in time like trigonometric terms depending on the longitude θ, storing them on arrays and reusing them during the simulation. Using built-it functions to compute complicated exponentials like those in the Manning formula. Although the optimizations provided speed up no sacrifice was incurred on precision. All GPU computations are performed on double precision.

### 5.1.1 Halo update on GPU

Update of the halo region of each mesh block after each time step with the latest values from neighbor blocks represents three different kinds of exchanges: copying, coarsening or interpolating. These operations are performed entirely on GPU. Kernels designed for each kind of exchange were created. In order to efficiently process the block edges, three lists are generated containing the list of halos that require each operation. This way the kernels can be launch concurrently and each focus on a different task minimizing the need for conditional divergences.

### 5.1.2 Specialized kernel types

By analyzing the domain's bathymetry it is easy to notice that some mesh blocks contain only wet points while others are a combination of dry and wet points. This idea is used to replicate the SSWE kernel in two variations.

The first SSWE kernel, named *Wet*, is used to compute the wave free propagation on wet-only blocks. The second SSWE kernel, named *Dry*, is used to compute the wave propagation with coastline boundaries in wet-dry mixed blocks. The main difference in the code between them being the additional treatment for the wall boundaries at coastlines in the case of the *Dry* kernel. A third kind of kernel called *Inundation*, specializes in computing the run-up on dry blocks inside focal areas.



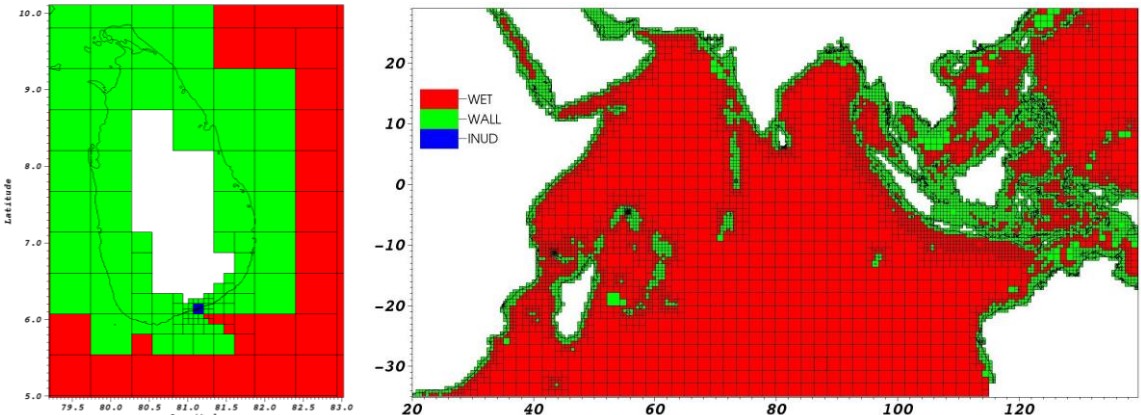

**Fig. 7 Mesh blocks colored by kernel type, Wet, Wall and Inundation. Left: zoom over Sri Lanka FA to highlight the inundation kernels shaded in blue. Right: Kernel type distribution on the entire Indian Domain.**

The result of the kernel assignment is illustrated Fig. 7 where blocks flagged as *Wet* are shaded in red, *Dry* blocks are shaded in green and *Inundation* blocks in blue. As expected *Dry* blocks tend to extend where coastlines lie while *Wet* blocks are spread out in the open ocean. When inside a focal area, dry-type blocks at level 7 are re-flagged as Inundation type. An example of this can be seen in the left image of Fig. 7 for the Sri Lanka FA with inundation blocks in blue. Whereas a single kernel would be too complicated and inefficient to compute the entire domain, splitting down the computation in specialized

kernels for each type of block not only provides a simpler way to process the blocks through lists but also gives the ability to fine tune them independently for higher performance.

## 5.2 Space filling curve and multi-GPU

In order to implement multi-GPU for further acceleration, first an appropriate domain partition must be chosen to guarantee an even work load among cards. Since a uniform mesh is not being used, this partition is non-trivial. Although block connectivity is kept using a *quadtree* structure, this does not provide information about the blocks ordering. For this purpose we use the space filling curve (SFC) (Sagan, 1994) as a way to trace the blocks ordering on the domain.

      SFC is a curve that fills up multi-dimensional spaces and map them into one dimension. It has many properties desirable

for domain partition, it is self-similar and it visits all blocks exactly once. We use the Hilbert curve in this work since it tends to preserve locality, keeping neighbors together and does not produce large jumps in the linearization like other curves tend to, such as the Morton curve. Figure 4 shows the Hilbert curve generation as a red line overlying the quadrants. It starts as a bracket on the first four quadrants, and with each spatial refinement, the bracket gets replicated subject to rotations and



reflections to guarantee the characteristic of the curve. The result of generating a Hilbert SFC for the Indian Ocean domain is shown in Fig. 8. By using this curve as a reference it is possible to establish the block ordering to partition the domain on even portions. The result of splitting the domain for 8 GPUS is shown in Fig. 9, where each portion is represented by a different color. In this case, 7 GPUs have a total of 981 blocks each, and the 8[th] a total of 982, making it a well balanced

5   partition. Different tests using 1, 2, 3 and 4 GPUs also achieved balanced partitions.

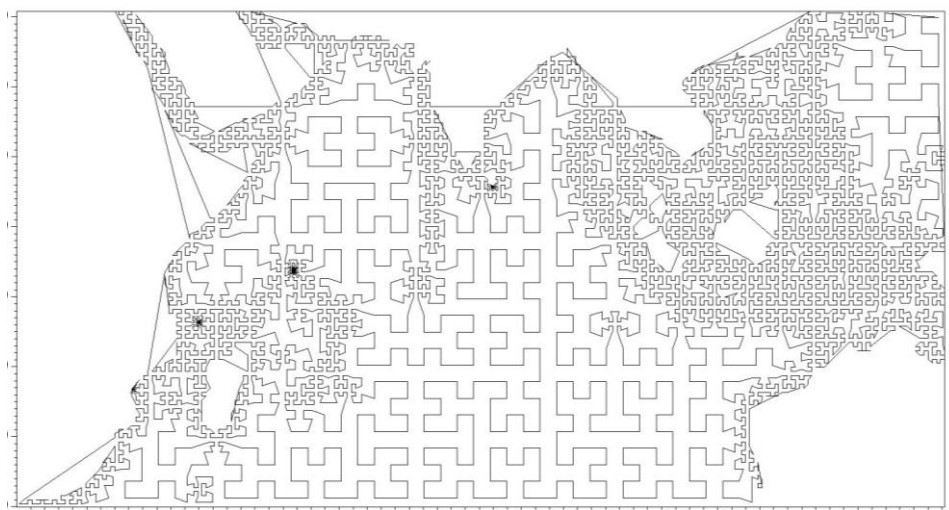

**Fig. 8 Hilbert Space Filling Curve for Indian Ocean Domain with four FAs**

Introducing multi-GPU also introduces the need of a buffer communication between cards. In the current CUDA GPU memory model, global memory cannot be accessed between different cards. This exchange is achieved by preparing buffers on GPU memory, downloading to CPU memory, using MPI to exchange the messages and uploading the received buffer to GPU memory.

15   In order to handle the communication structures and to produce buffers that do not represent a large communication overhead, we construct buffers following the *user datagram protocol* (UDP) (Reed, 1980) design, a concept traditionally used in network and cellular data communication. In this way, it is possible to eliminate the need for communication look-up tables while at the same time it makes the buffer exchange smooth and simplified. As depicted in Fig. 10, the first step consists on collecting all the halos to be transferred in a single buffer on GPU memory. This buffer is designed like in UDP,

20   with a header in front of every chunk of data. This header contains three bits of simple information: the destination block, the destination edge and the total size of its data. By including a simple 3-data header before the sent values, it is possible to organize the buffer in any way that packing/unpacking occurs smoothly and seamlessly.




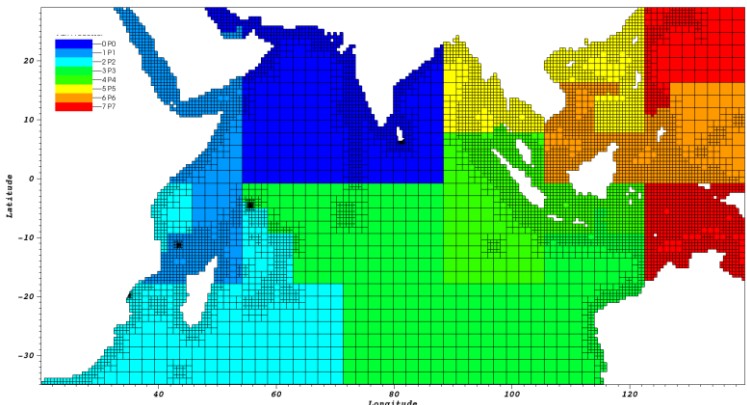

**Fig. 9 Indian Ocean Domain partition for load balance for 8GPUs. Each color represents a different GPU.**

By using this method no extra memory is needed to store communication tables or exchange them between processors. A single-buffer transfer between processes drastically reduces the communication time as opposed to transferring each halo individually.

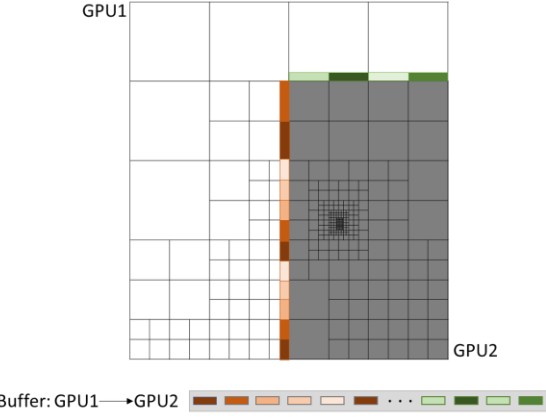

**Fig. 10 Multi-GPU communication. GPU buffer data collected and packed for a single communication.**





### 5.3 Variables and rendering output

The full work-flow of TRITON-G is depicted in Fig. 11, where the GPU flow is composed of two parts, the main simulation, which includes computing the fault source, wave propagation and inundation and the output compute and storage.

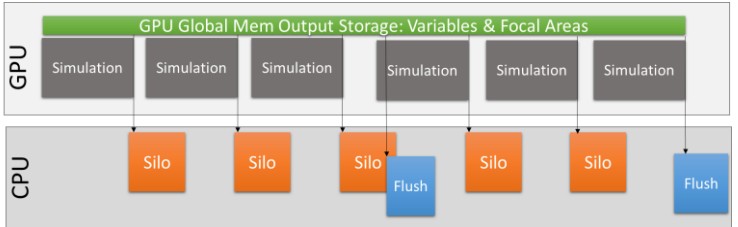

**Fig. 11 TRITON-G computational flow.**

For post-processing analysis purposes, output for the wave maximum height, maximum inundation, arrival time, flux and gauges is created. These are computed during simulation and stored on GPU memory, then flushed to CPU when
required by the user. A full-domain rendering at a regular frequency is also produced during simulation, while for the FAs, wave values at a much higher frequency are stored. These values are used for rendering at post-processing to avoid unnecessary output overhead.

TRITON-G generates SILO format files (Lawrence Livermore National Laboratory, 2017) filled with values from all blocks to generate the rendering images. Even though the image generation for the entire domain is not very frequent, the
process of generating a SILO file for such a large mesh represented a considerable overhead of around 15 to 20% of the total runtime. In order to minimize this unwanted effect we took advantage of the Piping mechanism. *Pipe* is a system call that creates a communication between two processes that run independently. In this way, a parent program can launch a child program and both run completely different tasks at the same time without interrupting each other. Using this concept, first a utility to create the SILO files for the entire domain was created as a stand-alone application. During execution, TRITON-G
calls this sub-program when a SILO file has to be written, running both simultaneously. Data between them is shared through the CPU shared memory. Figure 12 shows the advantage of implementing Pipe asynchronous output. Unlike traditional asynchronous output that relays on a large computational time to hide output, this Pipe method provides the ability to hide the output processing behind several computational time-steps. The result is an almost total elimination of the output overhead. Measurements showed that the output process after optimization represented just 1 to 2% of the total time,
practically removing the overhead.

The size of the output produced during simulation depends on user input parameters. For a 10-hour simulation with an output frequency of 4 minutes for the entire domain and 5 seconds for four FAs the required memory storage is around 65GB.

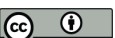



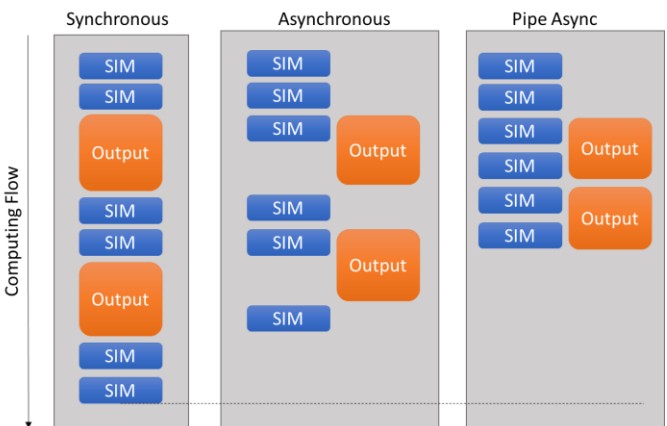

**Fig. 12 Output overlap and optimization using Pipes**

### 5.3.1 Sub-cycling implementation

5    A sub-cycling technique was introduce in order to increase the computational time step and speed the computation further up. Sub-cycling consists in setting a larger than the minimum time-step as a global time-step $\Delta t$, and making blocks with smaller local time-step cycle in sub-steps ($ns$) to match the global $\Delta t$. The time-step $\Delta t$ is calculated in each level using the Courant-Friedrichs-Lewy condition (CFL) (Courant et al., 1967). The CFL number is set to 0.80 for this work.

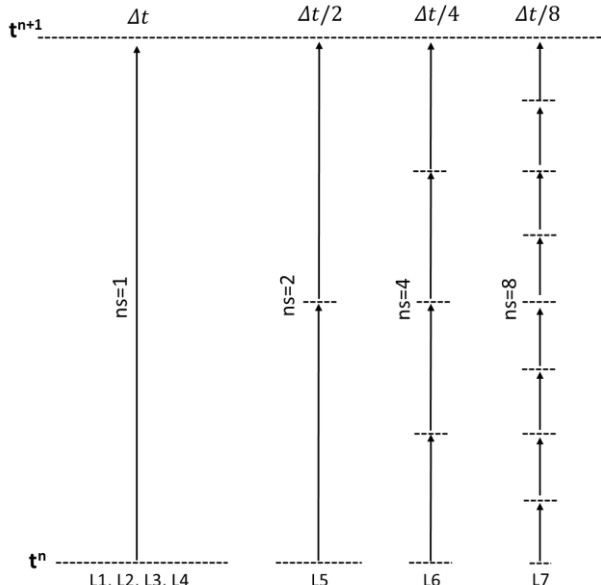

**Fig. 13 Illustration of the sub-cycling process for each level.**



A graphical illustration of the sub-cycling implementation is shown in Fig. 13. As reference, a block at level 5 in this sub-cycling technique has a time step of Δt /2, which implies that it requires two cycles to match the global Δt. Blocks with the same number of sub-cycles (Levels L1, L2, L3 and L4) are grouped in a single list. While in theory the larger time step brings up speed, the potential downside is that too many blocks sub-cycling can create a large work-load overhead resulting in a slow-down of the whole computation. To avoid this, a global Δt of 1.6s is chosen to sub-cycle only blocks with levels over level 3. The reason being, that around 80% of the total number of mesh blocks are level 3 and sub-cycling them would represent too large an overhead and would defeat the whole purpose of applying this technique.

In general after a large Δt step, corresponding boundary conditions are interpolated in time to update the sub steps. However this procedure introduces an additional computational overhead. To pursue the fastest modelling possible, TRITON-G rescinds the boundary generation and instead uses the available boundary values at time $n$. Based on the benchmark and hindcast comparison, this decision proved to be acceptable based on the good agreement and accuracy of the results.

Introducing this sub-cycling technique varies the GPU load initially created since a single block might be computed more than once. In order to guarantee the load balance, an additional weight is applied to the space filling curve. Each block gets attributed a *weight* during the SFC generation equal to the number of sub-cycles it requires. This approach for the domain partition allows to create a fair work re-balance on the GPUs. Implementation of this sub-cycling technique showed a speed up of around 15% in the total wall clock runtime.

### 5.3.2 Runtime performance

Several tests to estimate the performance of TRITON-G were done. Results ran on the Supercomputer *Tsubame* 3.0 (Tsubame, 2017) are presented, with Intel Xeon E5-2680 2.4GHz × 2, RAM 256GB, NVIDIA Tesla P100 (16GB) × 4/node, CUDA 8.0, gcc 4.8.5 and Openmpi 2.1.1.

As comparison, results on a second machine are also presented, using Tesla K80 (12GB×2) cards, with Intel Xeon CPU E5-2640 @2.6 GHz, RAM 128GB, CUDA 8.0, gcc 4.7.7 and Openmpi 1.8.6.

The breakdown of the main parts of the simulation using 3 GPUs is shown in Fig. 14, where *Inund* stands for Inundation kernel, *Wall* stands for the wall kernel, *Wet* for the *Wet* kernel and *X* and *Y* for the direction of the computation equivalent to longitude and latitude respectively. The process of updating the halos, presented in the graph as *Bnd* represent only 9% of the total running time. It can be seen that the Wet and Wall kernel have similar performance despite the fact that the wall includes additional treatment for the coast boundaries. Since this treatment consists of many conditionals and they were replaced during optimization, it is understandable that the performance is similar. The slide *Others* include several values, most importantly communications which represents around 1.5-2.0% of the total running time.





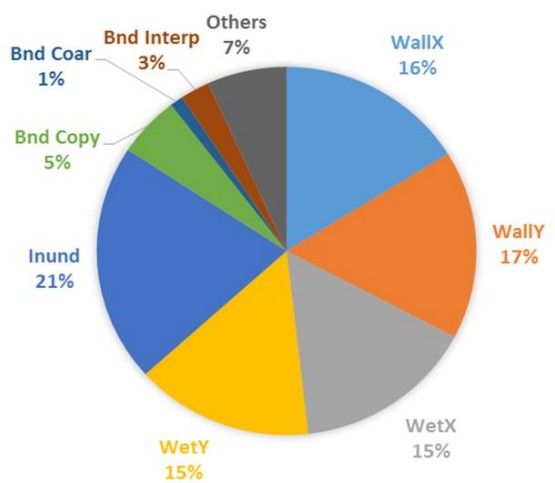

**Fig. 14 Computing breakdown shown in percentage**

Results for runtimes using Tesla P100 cards and Tesla K80 cards are presented in Fig. 15 for 1, 2, 4 and 8 GPUs. For this
5   test, 10 hours were simulated on the mesh initially generated for the Indian Ocean Domain (Fig. 5). A comparison between
both GPU cards shows a speed up of almost 4 times from the older K80 cards to the latest P100 on Tsubame 3.0. In our
collaboration project with RIMES an objective to complete this test under 15 minutes was set, which could be fulfilled by
using 3 to 8 GPUs in this configuration.

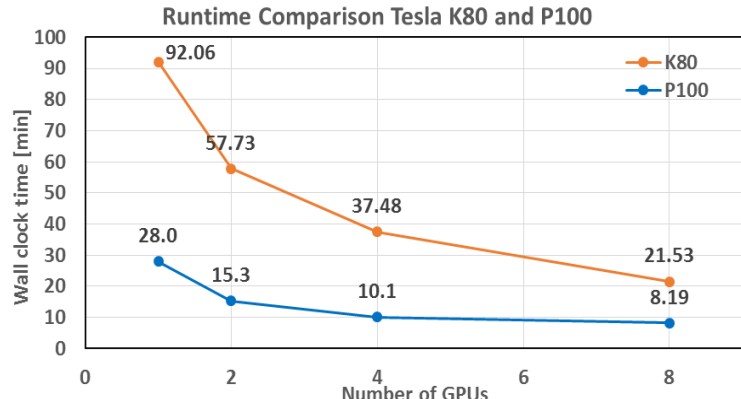

**Fig. 15 Wall clock comparison of 10-hour simulation on Tesla K80 and Tesla P100**



A saturation is noticeable in Fig. 15 as the number of GPUs are increased. A possible reason for this phenomenon is related to the increase of buffer preparation, packing/unpacking, and the communication exchange. Using the same domain size for all cases is another possible reason. Having fewer blocks on each GPU generates lower occupancy which might degrade performance. However, having met this study's time-to-solution objective of less than 15 minutes, no further

optimization was deemed necessary.

By measuring the time required for the first wave to arrive in the focal areas, it was found that for Sri Lanka. Using 4 GPUs just 2 minutes wall clock time is required to generate the results of the inundation. The real tsunami wave took approximately 2 hours to propagate from the initial source to Sri Lanka, obtaining simulation results faster than real time gives authorities sufficient time to make decisions regarding evacuations.

**6 Case study**

In order to compare and validate the results of TRITON-G under a real tsunami scenario we use the hindcast of the 2004 Indonesia tsunami. Results for propagation, gauges and inundation comparison are represented. A preliminary test using the tsunami propagation over a submerged circular shoal benchmark proposed by (Yoon et al., 2007) was used for validation. In this test, a circular shoal is submerged in a domain of 1500km × 500km with a solitary wave input as initial condition.

Without the effect of the dispersion, comparison of TRITON-G results with the reported gauge values show good agreement in arrival time and wave height, which served as a validation of the ability to transport a wave train over long distances without it being diffused.

**6.1 Indonesia 2004 tsunami hindcast**

This event occurred at 7:58 am on December 26$^{th}$, 2004, with a magnitude of 9.0 Mw generated by the subduction of the Indian Plate by the Burma plate. Nearly 1600 km of fault was affected around 160km off the coast of Sumatra (Titov et al., 2005). This massive earthquake generated a large tsunami that spread over the Indian Ocean in the following hours.

The tsunami wave propagation computed by TRITON-G is depicted in Fig. 16. Each subsequent snapshot represents three hours after the earthquake's main event. A synoptic qualitative comparison with existing field surveys and simulations confirmed a correct propagation of the initial wave train, however to check the validity of the results, two kind of comparison are presented for tide gauge records and for inundation map simulations.





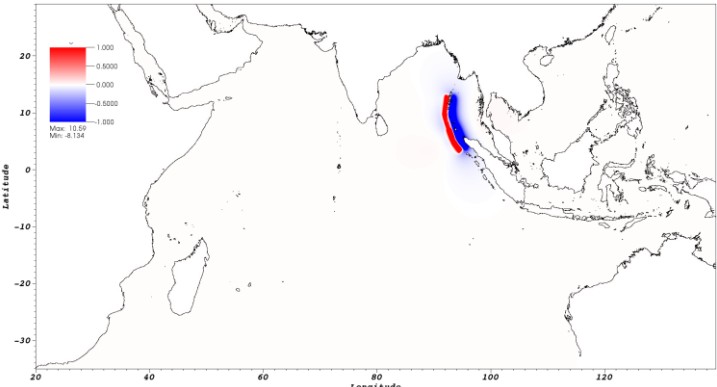

**Fig. 16 (a) Time = 0 hours. Initial Source in Sumatra.**

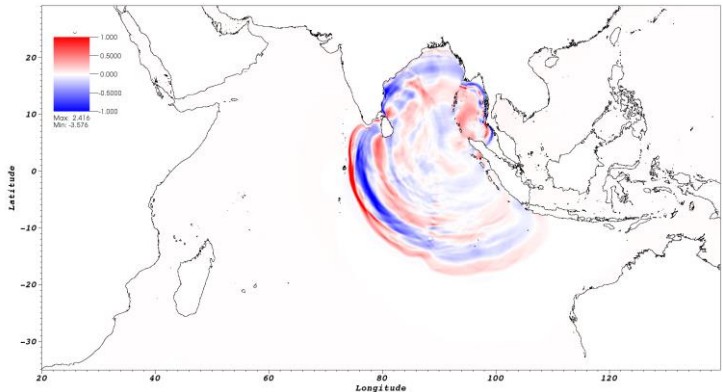

5                                             **Fig. 16 (b) Time = 3 hours.**

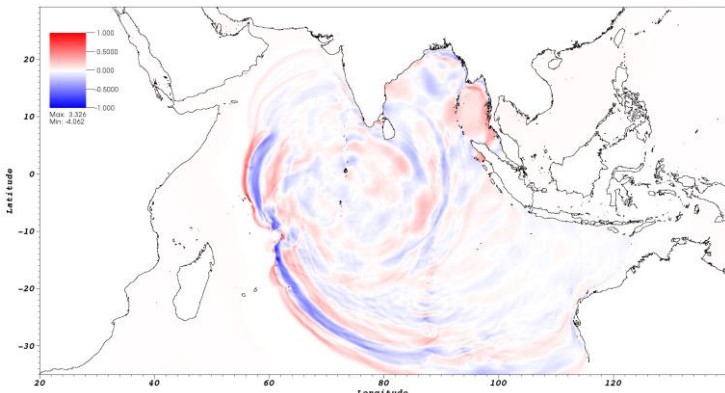

**Fig. 16 (c) Time = 6 hours.**

**Fig. 16 Snapshots every three hours (a) – (c) of the Indonesian 2004 tsunami propagation simulated by TRITON G**


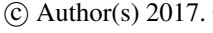


**6.1.1 Tide gauges comparison**

To check the correctness of the wave propagation, buoys located in different parts of the Indian Ocean were used to compare

5 TRITON-G results. These buoys measure the ocean sea level at regular intervals and serve as a critical factor to determine tsunami wave arrival times and heights. Gauges recorded at the moment of this event were obtained from NOAA's tsunami events database and inundation maps were obtained through RIMES. Results from RIMES previous operational model are also included for comparison. Their previous model was based on a customization of TUNAMI (Srivihoka et al., 2014) to include four nested grids with fixed resolution of 2 arc-min, 15 arc-second, 5 arc-second and 5/3 arc-second.

10    Results for five stations are shown. *Diego Garcia* Fig. 17(a) in an atoll in the Chagos Archipelago, located at 7º30'N 72º 38' E. *Male* Fig. 17(b) near the Maldives Islands, located at 4º18'N 73º 52' E. *Gan* Fig. 17 (c) near the Maldives Islands, located at 0º68'N 73º 17' E. *Colombo* Fig. 17 (d) in Sri Lanka, located at 64º93'N 79º83' E. *Point La Rue* Fig. 17 (e) near Seychelles, located at 4º68'S 55º53' E.

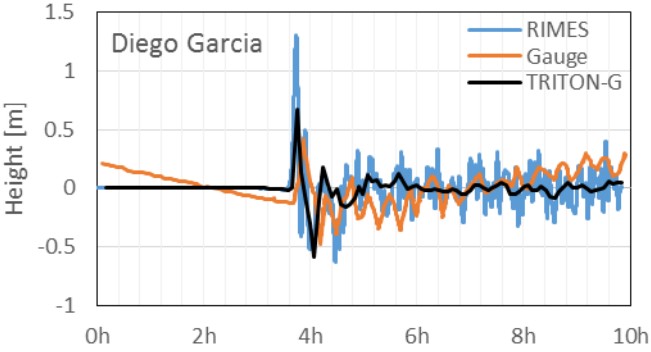

**Fig. 17 (a) Comparison of arrival wave at Diego Garcia, tide gauge and model results**

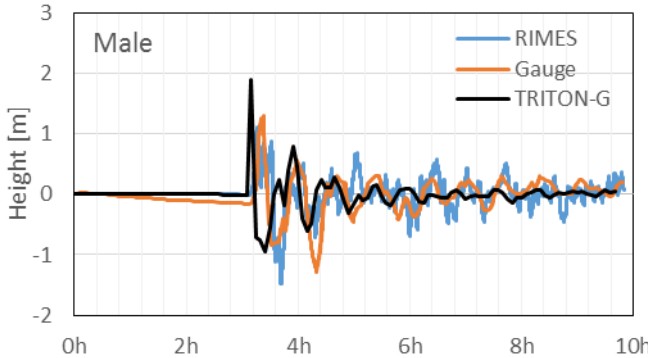

**Fig. 17 (b) Comparison of arrival wave at Male, tide gauge and model results**



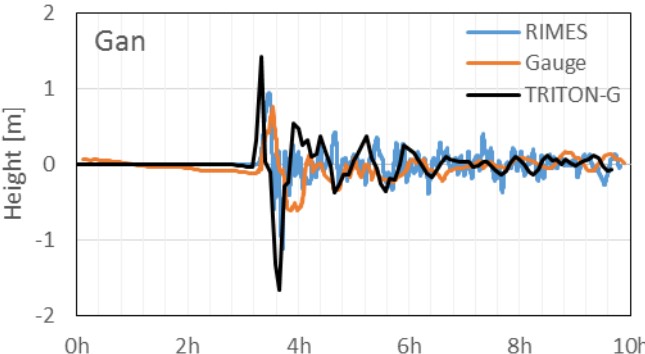

**Fig. 17 (c) Comparison of arrival wave at Gan, tide gauge and model results**.

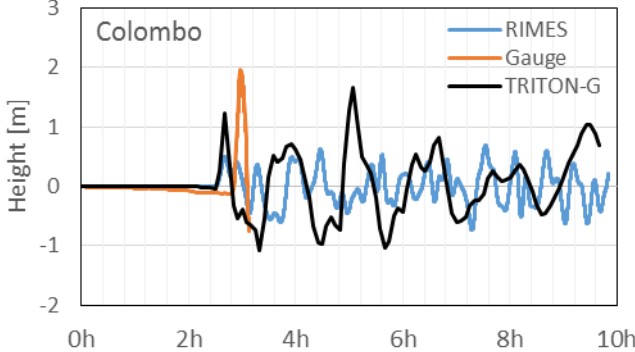

5          **Fig. 17 (d) Comparison of arrival wave at Colombo, tide gauge and model results.**

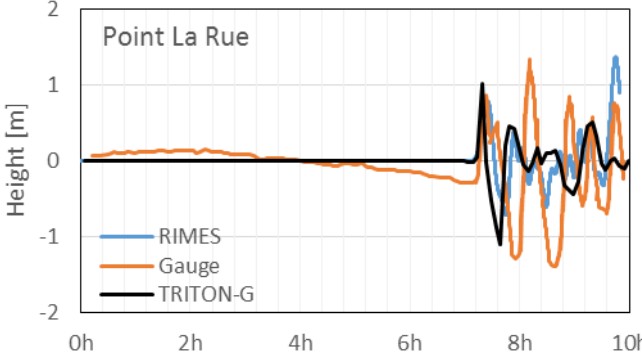

**Fig. 17 (e) Comparison of arrival wave at Point La Rue, tide gauge and model results.**

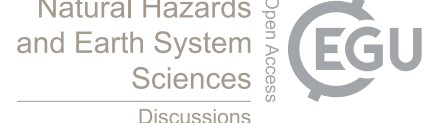


The comparison between the tide gauges TRITON-G and RIMES's model based on TUNAMI are shown in Fig. 17. As it can be seen, the arrival times are in good agreement with the measured ones. The main event peaks are also reproduced in all cases with the crests' signs are in accordance with the measured values. The effect of tide is not considered in the current model which explains the height differences at initial times in the results. In the case of Male, three of the first peaks were

also estimated in the simulation. The case of Diego Garcia serves also as a test for long propagation, since it is located around 2700km and there is no topography between source and station. This makes it a good way to validate that the wave is properly propagated at the right speed and no effects of diffusion on the wave height are present. Diego Garcia and Colombo (which recorded only around 3 hours before being damaged) are two examples of obtaining a more accurate and closer results than the previous model used at RIMES, where a closer height to the measured peak was obtained. Point La Rue

represents also a good test for long propagation for TRITON-G numerical model since the location is over 4500km from the source and the wave has travelled over complex bathymetry and reflected on multiple coastlines. However the arrival time is still in good agreement as well as the wave arrival peak height. No effect of wave main peak diffusion is noticeable.

The time arrival differences of a few minutes between measurement and TRITON-G simulation can be partly explained by the location of simulated gauge. Even though the main events could be reproduced, a tendency to overshoot is noticed,

nonetheless this did not affect the ability of the model to transport the wave along far distances and in no case an arrival wave sign was reported incorrectly. We discuss briefly three main reasons for the difference in arrival height and wave oscillation after the main event. The first is related to bathymetry and topography. Although databases for bathymetry and topography with good accuracy are available, these are still far from representing in detail the real shape of the ocean's bottom and topography. This difference makes it challenging to reproduce the wave reflections on coasts and effects of

traveling through the ocean bottom completely realistic. Based on this, it is expected that some differences are found in the wave reflections and oscillations. A study about the influence on bathymetry resolution can be found in (Plant et al., 2009). The second reason relates to the dependence of every tsunami model on a good and accurate initial condition to obtain good simulations. The use of inaccurate initial fault source can affect the resulting simulation especially in locations near the source. This is particularly challenging since it is not possible to measure precisely the ocean surface at the moment of a

tsunami event. The third reason is related to dispersion. Waves traveling through the Ocean bottom experience physical dispersion due to the effect of the bathymetry. In general, this dispersion is compensated by numerical dispersion introduced by the truncation error. However TRITON-G utilizes a cubic interpolation upwind scheme that has the advantage of minimizing dispersion and diffusion. An almost homogeneous traveling train wave with minimum dispersion effect is produced instead, reducing the possibility of seeing the higher oscillatory behavior of the arrival tsunami wave seen in the

gauges. These kinds of discrepancies had been observed and reported on several other operational models as well (Dao and Tkalich, 2007), (Grilli et al., 2007) or (Arcas and Titov, 2006).



### 6.1.2 Inundation maps comparison

A further validation for TRITON-G model is to compute inundation in certain areas and compare it with field surveys or existing maps. Since inundation maps that are exactly measured do not exist, we present comparisons with RIMES' existing

5  simulated inundation maps (RIMES, 2014) and post-tsunami field surveys. Two cases are presented, the first in Hambantota (Sri Lanka) and the second in Phuket (Thailand).

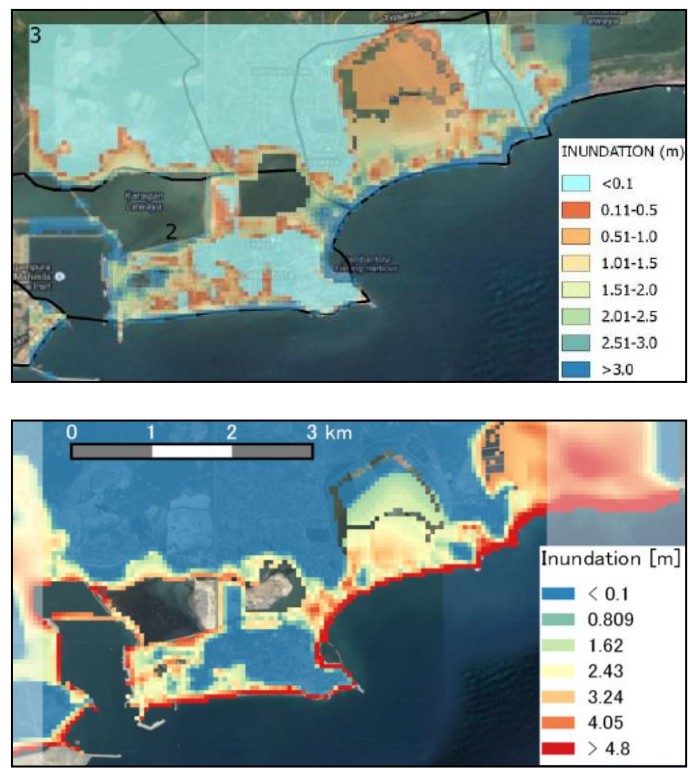

10  **Fig. 18 Inundation comparison for Hambantota, Sri Lanka. Top: RIMES model. Bottom: TRITON G model.**

The first inundation validation presented is the result for Hambantota in Sri Lanka. Eye-witness accounts report the arrival time of the first tsunami wave around 9 am the morning of the 26th, some two hours after the initial earthquake in Sumatra. This coincides with TRITON-G's predicted arrival time of two hours for this region. The inundation map for

15  Hambantota generated by TRITON-G is shown in Fig. 18 bottom. According to measurements done post-tsunami, it was determined that the arrival waves had heights of over 8 meters and produced run-ups inland in certain areas of up to 2 km.

TRITON-G inundation result also shows areas up the coastal bay where run-up produced hundreds of meters deep run-ups in land, coinciding with the recounts. By comparing it with the result provided by RIMES we found that both





simulations show agreement with each other on the areas that experienced and did not experience inundation. The decisive factor that made some areas more prone to inundation than others was the topography. The arrival tsunami wave hit the coast with heights of around 8-10 meters. Coastal areas that faced the ocean with higher topographic heights were spared from being inundated. On the contrary, coast shores that were practically flat were overtaken by the incoming wave as shown in

5    the results.

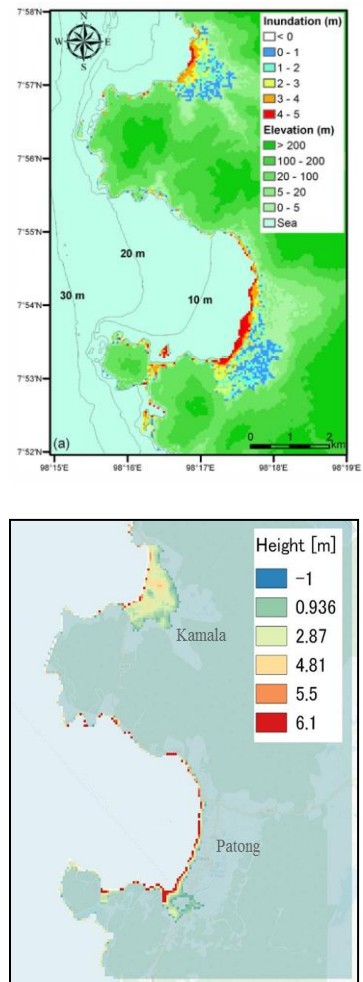

**Fig. 19 Kamala (North) and Patong (South) inundation maps comparison. Top: inundation result by (Supparsri et al.,**

10   **2011). Bottom: TRITON G inundation result.**

Results for the second inundation validation in Phuket are compared with those of (Supparsri et al., 2011). The wave arrival time for this region is of around 181 minutes, which agrees with the values obtained by TRITON-G model of





180 min. Inundation results are shown in Fig. 19, the image on top presents the inundation simulation obtained in the report while the image on the bottom depicts the results of TRITON-G model.

The results around the Kamala region coincide very well between models. Both report maximum inundation heights of around 5-6 meters and the run-up distances follow the same pattern. In the south, at Patong region however there is a

difference in the run-up distances. This is explained by the difference in the bathymetry used by TRITON-G. While in the (Supparsri et al., 2011) study a 52m resolution was used on the entire inundation area, in our model 50m resolution bathymetry was available only in Kamala. For Patong, values were interpolated from a lower 150m resolution database which produced a smoother topography and less accurate run-up results. This test served to validate the ability of TRITON-G to estimate inundation but also highlights the importance and the effect of having accurate and realistic

bathymetry for the simulation.

**7 Conclusions**

The tragic events of recent tsunamis showed the importance of developing fast and accurate forecasting models. We implemented several techniques to reduce the time-to-solution to meet our runtime goals in the successful development of

this fast and accurate tsunami operational real-time model. In a short time, wide-area simulations (ocean size) can be obtained much faster than real time, meeting our goal for results in less than 15 minutes. The combination of highly accurate numerical methods with light stencils provided an excellent solution to the governing equations, and gave stability on complex bathymetry. A customized, tree-based refinement that captured complex coastline shapes was successfully implemented using two factors; distance and focal areas. Using the distance from the coast to refine allowed to leave coarser

blocks in the open ocean while blocks near the shoreline were refined to a higher 50m resolution. Focal areas were also successfully introduced in the refinement to delimit the regions where the high-resolution blocks were generated, and to use memory and computational resources efficiently. A full-GPU double precision implementation was proven successful in delivering a large speed up. All parts of this simulation, including output storage are processed entirely on GPU with specialized kernels. For multi-GPU, the use of a weighted Hilbert space filling curve successfully generate balanced domain

partitions and work load.

Using Tsubame 3.0's GPU Tesla P100 cards for a full scale simulation of 10 hours resulted on a wall clock time of just under 10 minutes with 3 GPU cards, including considerably-sized output (65GB) and using double precision. The hindcast of the Indonesian 2004 tsunami served to compare and validate TRITON-G simulation results, finding very good agreement with gauge propagation and inundations. The flexibility and robustness of TRITON-G allows it to be an excellent

operational model that can be easily adjusted for different tsunami scenarios, and its speed permits it to be a real-time forecasting tool. For these reasons, and under the collaboration with RIMES, TRITON-G has been successfully deployed as their operational model since August 2017.



**Data availability**

Underlying research data can be found in Open Science Framework repository:

Acuna, M. A., Takayuki, A.: TRITON-G, [online] Available from: osf.io/fydz8, 2017.

**Competing interests**

The authors declare that they have no conflict of interest.

**Acknowledgements**

This work has been supported by KAKENHI Grant-in-Aid for Scientific Research (S) 26220002 from the Ministry of Education, Culture, Sports, Science and Technology (MEXT) of Japan and the Japan Science and Technology Agency (JST) Core Research of Evolutional Science and Technology (CREST) research program "Highly Productive, High Performance

Application Frameworks for Post Peta-scale Computing". The authors thank Prof. Kiyoshi Honda, Chubu University and the staff of RIMES (The Regional Integrated Multi-Hazard Early Warning System for Africa and Asia) for their extensive support and the Global Scientific Information and Computing Center at Tokyo Institute of Technology for the use of their supercomputers TSUBAME2.5 and 3.0.

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
