# Peer review of "Tree-based mesh-refinement GPU accelerated tsunami simulator for real time operation"

_Natural Hazards and Earth System Sciences, 2017_

## Referee Comment (RC1) · Anonymous Referee #1 · 26 Jan 2018

Overview

In the current high-performance computing (HPC) context, where accelerated computing is revolutionizing not only the computation capacity but also the computation time required to perform the numerical simulation of large problems, it's very important the development of new numerical models able to change the game rules in the context of natural hazard early warning. In this framework these authors develop a new tsunami model (TRITON-G) coded in GPU architecture. The model is based on Spherical non-linear shallow water equations (SSWE) in the far-field area while the near-field and inundation areas the cartesian coordinate version of SWE is used. The developed nu-

merical model in the far-field is based on the method of characteristics with a cubic polynomial approximation on the grid. In the near-field area, when inundation is calculated, they use a finite volume model. Authors use a quad-tree refinement method to build the grid for the computational domain. Later, it's explained in detail the GPU and multi-GPU implementation and finally they present some numerical experiments based on analytical solutions and a simulation of the Indian ocean 2004 tsunami. The article closes with final conclusions and a list of 67 references.

Overall Recommendation

My recommendation is: major revision.

Assessment and Further comments

Included in the PDF file attached.

Please also note the supplement to this comment:
https://www.nat-hazards-earth-syst-sci-discuss.net/nhess-2017-379/nhess-2017-379-RC1-supplement.pdf

**Supplement:**

**Review of: Tree based mesh-refinement GPU accelerated tsunami simulator for real time operation**

Marlon Arce Acuña, Takayuki Aoki

**Overview**

In the current high-performance computing (HPC) context, where accelerated computing is revolutionizing not only the computation capacity but also the computation time required to perform the numerical simulation of large problems, it's very important the development of new numerical models able to change the game rules in the context of natural hazard early warning. In this framework these authors develop a new tsunami model (TRITON-G) coded in GPU architecture. The model is based on Spherical non-linear shallow water equations (SSWE) in the far-field area while the near-field and inundation areas the cartesian coordinate version of SWE is used. The developed numerical model in the far-field is based on the method of characteristics with a cubic polynomial approximation on the grid. In the near-field area, when inundation is calculated, they use a finite volume model. Authors use a quad-tree refinement method to build the grid for the computational domain. Later, it's explained in detail the GPU and multi-GPU implementation and finally they present some numerical experiments based on analytical solutions and a simulation of the Indian ocean 2004 tsunami. The article closes with final conclusions and a list of 67 references.

**Overall Recommendation**

My recommendation is: major revision.

**Assessment and Further comments**

In the introduction, authors take a tour of the different tsunami forecasting models by classifying them in two groups: depth-averaged non-hydrostatic and hydrostatic models. Although authors discuss briefly about some well-known in the tsunami context models, I don't understand the criteria followed for leaving out some important models, especially those models used by different Tsunami Warning Centers along the world: that is the case of COMCOT (http://223.4.213.26/archive/tsunami/cornell/comcot.htm), NEOWAVE (Non-hydrostatic Evolution of Ocean WAVE) or even in the GPU framework, Tsunami-HySEA (https://edanya.uma.es/hysea/index.php/2015-10-08-15-52-45/tsunami-hysea) or more recently the GPU version of NAMI DANCE (http://namidance.ce.metu.edu.tr/). Of course, it's not possible to include all the developed tsunami models, but as the development of TRITON-G was focused on its operational capabilities, it would be expectable to focus this analysis on models used in

this context, moreover if some these models have been developed in the GPU framework (for example: the MOST model -included in the paper-, but that has been fully re-coded in GPU-, or Tsunami-HySEA that is being used by the Centro di Allerta di Tsunami (CAT) by the INGV (Italy) in the NEAMTWS context or the Joint Research Center of the European Union through the Global Disasters Alert and Coordination System (GDACS) (http://meetingorganizer.copernicus.org/EGU2015/EGU2015-13797-3.pdf)

**Section 2**

In section 2, authors describe the spherical non-linear SWE used in the tsunami-propagation and refer to Toro, 2010 for the cartesian coordinate version of the SWE used when inundation needs to be computed. This election will mean the use of different numerical methodology (described in section 3) depending on the area where computations are performed. So, regarding the numerical methods used in this paper: in the propagation-stage it's used the method of characteristics (MOC) in combination with a cubic-polynomial interpolation to find the interpolated values on the grid. Authors recall that the numerical method described is well-balanced but on other hand I'm missing two other important features that should be proven: conservation of mass and water height positivity. Is the total mass conserved? For instance, what about the mass conservation in the experiment described in p8 line 10? Regarding the water height positivity ($h$), is it guaranteed in the propagation stage even considering the cubic-polynomial interpolation process?

On other hand, what is the convergence order of the numerical models used on each area?

**Section 3**

As the coordinate system used combines spherical coordinates and cartesian coordinates, how is treated the boundary between the considered domains on each coordinate system?

Regarding the run-up calculation, in section 3.2 authors propose a technic based on considering ($hu=hv=0$) when $h$ is less than a certain small fixed quantity. They confirm that the proposed implementation has been proven to be robust and stable under different benchmarks. Please, include a reference where this numerical technic is used.

Finally, in this section it's included a parabolic bowl problem as validation problem. That's a good synthetic test but I consider that it's not enough to define a numerical model as validated when such numerical model that is going to be used with real topo-bathymetries in real cases, moreover when, as it's remarked by authors, the model is going to use in the RIMES context. In order to validate this numerical scheme to be used in real cases I suggest, for instance, the use of the inundation benchmark experiments proposed by the National Tsunami Hazard Mitigation Program (NTHMP) in http://nws.weather.gov/nthmp/documents/nthmpWorkshopProcMerged.pdf or [Horrillo, J., Grilli, S.T., Nicolsky, D. et al. Pure Appl. Geophys. (2015) 172: 869. https://doi.org/10.1007/s00024-014-0891-y] or [Macías, J., Castro, M.J., Ortega, S. et al.

Pure Appl. Geophys. (2017) 174: 3147. https://doi.org/10.1007/s00024-017-1583-1] where problems with analytical solutions, laboratory experiments and real problems with field measurements are proposed. I would suggest studying the behavior of TRITON-G at least in the "mandatory" benchmark problems: BP1, BP4, BP6, BP7 and BP9.

Regarding the field measurement experiments, authors show in section 6.1.2 some inundation maps in different locations (Fig. 18 and 19) where the comparisons are made basically against other models. I think that it cannot be considered a numerical model as validated with field measurements by making comparisons basically with other models results. There are available many tsunami field measurements to validate the inundation process. In this sense you can consider cases where there are more available data than in the case studied in section 6.1.2. For instance: BP7, BP9 or many inundation scenarios related to the Tohoku, 2011 event where detailed data are available.

Regarding the generation process described in section 3.3, authors use the coseismic deformation proposed by Smylie and Manshiha, 1971. I'm surprised at this point because in [Okada, Y. (1985), Surface deformation due to shear and tensile faults in a half-space, Bulletin of the Seismological Society of America, 75(4), 1135-1154] this work is extended, and it's provided also an analytical way to get the "Okada model" coseismic deformation that can easily be extended to CUDA in the GPU context. I would like to know the reason for not considering Okada as model for the generation process.

**Section 4**

In section 4, authors use a tree-based mesh refinement similar to the AMR technic used by R.J. LeVeque in GeoClaw. In this case the refinement is not made automatically as in GeoClaw, but it's customized to be refined in coastal areas and focal areas. It's an interesting alternative to the use of nested meshes when more detailed information is required in certain coastal areas. In p14, line 9-10 it is discussed the number of blocks necessary under the considered resolution and according the 65 x 65 node-centered cells if the number of blocks is 230,000 you would have 971,750,000 node-centered cells. How many information is stored on each node-centered cell (in double precision) to represent over 100GB of memory space?

**Section 5**

Regarding section 5, I have some comments, but my main concern is that I think that this section is out of the scope of this journal and my recommendation would be to publish it in a journal more related to this field. Anyway, I will make some comments regarding this section. To my knowledge the overall GPU implementation has been solved in a very efficient way, particularly the implementation of the pipe asynchronous output that is crucial to deal with the GPU-CPU traditional bottleneck. In p23, line 8 it is showed that you use CFL 0.8, is not stable the implemented model for CLF nearer to 1? On other way, in p24, line 5 it is used $\Delta t$ = 1.6 for blocks with levels over 3. Is this consistent with the

CFL condition? Or, can you ensure the stability of the numerical scheme under this assumption?

In section 5.3.2 it is showed the runtime performance by comparing results obtained with two different GPU architectures: Tesla K80 and Tesla P100. It must be remarked that the configuration of the computation nodes that you are using is different for each architecture. While the Tesla P100 nodes have 4 GPU's per node, the Tesla K80 nodes have 2 GPU's per node. Unless your tests are using only two P100 per node, the network communications will reduce the performance between the K80 in front of P100. What kind of network is being used in these clusters? Another point is, are output data being stored into hard disks in these tests or are they related only to pure computation time?

**Section 6**

Finally, my main comments about section 6 have been included in my recommendations for section 3. Anyway, I have some specific comments about this section. It would be nice if the color scale used in Fig. 18 and 19 are the same for each subplot. With these graphics we can compare the inundation extensions, but it's difficult to compare the inundation height when different scales are used on each graphic. On other hand, and as I pointed before, this test is not enough to say that the model is validated, so I don't agree with the sentence of p33, lines 8-9.

---

## Referee Comment (RC2) · Anonymous Referee #2 · 11 Feb 2018

** General Comments **

The paper presents an operational model for the fast simulation of the generation, propagation and inundation of tsunamis in wide areas by exploiting modern multiGPU hardware. The model is tested and compared under a real tsunami scenario, obtaining a nice performance results from the operational point of view. The implementation of this operational model on a cluster of multiGPU computers involves the suitable integration of numerical schemes (MOC with dimensional splitting to solve spherical SWE for the Tsunami propagation, Surface Gradient Method to solve the cartesian SWE for inundation areas, ...) and computing techniques (quadtree-based mesh refinement to

save resources, Hilbert Space-filling curves to preserve locality in the parallel partitioning, CUDA for GPU programming and MPI for remote communication, overlapping the computation in GPU and the generation of output files and rendering in CPU, etc.) to obtain an efficient complete CPU-multi-GPU operational model for Tsunami forecasting.This model would make it possible a very fast simulation which can help in the early identification of the tsunami consequences.

In my opinion, the techniques which are presented and the scientific data which are included are coherent and relevant and can be useful to scientists working in this area because all the approaches and techniques are devised in conjunction to perform very quickly realistic simulations. Although the paper is well organized and written, the reading of several pieces of the sections which explain the multiGPU implementation is not easy to understand and several implementation decisions which are presented are not clear. Moreover, in Section 5, I think that the use of technical and english language should be checked (several corrections are included in the Section of Technical Corrections).

On the other hand, to intend the validation of the operational model with a real tsunami scenario when the input data are not sufficiently accurate is very ambitious.

** Specific Comments **

In Section 1, it would be interesting to include a comparison with previous works related with the multiGPU simulation of tsunamis to obtain faster-than-real-time results.

In Section 5.1., the description of the configuration of the main CUDA kernels (second paragraph of the section 5.1.) is not easy to understand. A graphical description of the configuration and a description of the calculations assigned to each CUDA thread (relating this section with section 3 and 4) would be very useful to understand it.

The specialized kernel types presented in section 5.1.2. could affect the load balancing between GPUs because the computational execution cost of each kernel type on

a mesh block would be possibly different. I do not know if this fact is taken into account for the considerations included in section 5.2. I think it would be interesting to report graphically execution times for each particular GPU in order to evaluate the effectiveness of the domain partitioning and even to rethink the approach by designing a dynamic load balancer.

In Section 5.3.2., the configuration for the network which interconnects the TeslaP100-based nodes and the Tesla K80 nodes should be included to analyze Fig. 15.

In Section 5.3.2. and in the Conclusions, authors underline evidences about the wall clock time and the speedup which are obtained with 3 GPUs. However, the particular performance results for 3 GPUs are not reported and they are not included in Figure 15.

In Section 5.3.2., absolute performance measures on one GPU for the main kernels (it can be obtained by using the Nvidia CUDA profiler) could be useful to evaluate the efficiency of the CUDA implementation.

In section 6.1.2., the figure 18 Top which presents the RIMES results for Hambantota Inundation is not introduced textually in page 31, line 18. On the other hand, it is difficult to compare visually the RIMES and TRITON G inundation maps if the colours are used in a completely different manner for each map.

** Technical corrections **

- Page 13, Section 4.1, line 7: "blocks close to the coast until reaching a target ..."

- Page 15, Line 7,8: The sentence is not clear.

- Page 17, Line 20; The sentence is not easy to understand.

- Page 17, Line 21: "organized in a grid of blocks of CUDA threads ..."

- Page 18, Line 8: "branch diversion" is not the usual term. "branch divergence" is more frequent in this context.

- Page 18, Line 12: A comma in the sentence after "speed up" would help in order for it to make sense.

- Page 18, Line 18: "... This way the kernels can be laubched ...".

- Page 19, Line 5: "... is illustrated in Fig. 7 ...".

- Page 23, Line 5: "... was introduced in order ..."

---

## Author Comment (AC2) · 28 Jun 2018

Dear Referee, Thank you for your kind and useful comments to improve our work. Please refer to supplement PDF for full text responses along with new figures and tables.

Please also note the supplement to this comment: https://www.nat-hazards-earth-syst-sci-discuss.net/nhess-2017-379/nhess-2017-379-AC2-supplement.pdf

2017-379, 2017.

---

## Author Response (AR1)

**Response to Reviewer #1**

We would like to thank the reviewer for their precious time and for the thorough review and the many helpful comments and suggestions made to improve the present work. Please find below the reviewer's comments and author's replies to these comments.

Assessment and Further comments

Q1. "[…] I don't understand the criteria followed for leaving out some important models, especially those models used by different Tsunami Warning Centers along the World […]". "[…] Of course, it's not possible to include all the developed tsunami models, but as the development of TRITON-G was focused on its operational capabilities, it would be expectable to focus this analysis on models used in this context, moreover if some these models have been developed in the GPU […]"

A1. We based the previous existing work on an historical development and on the models we considered well known and that we were more familiar with, as well as some models mentioned in the National Tsunami Hazard Mitigation Program (NTHMP, 2012). As mentioned by the reviewer is not possible to include all models and we appreciate the important references provided about these other models left out which will be included in the introduction, specially the GPU models suggested since they are relevant to our research (p.2, at lines 8,13, 16 and 27).

Section 2

Q2. So, regarding the numerical methods used in this paper: in the propagation stage it's used the method of characteristics (MOC) in combination with a cubic-polynomial interpolation to find the interpolated values on the "[…] Is the total mass conserved? For instance, what about the mass conservation in the experiment described in p8 line 10?"

A2. On this regards, the discretization used for the run-up method is non-conservative. However, the difference from the initial mass and final mass is negligible and well below the 5% criteria that the NOAA Technical Memorandum OAR PMEL-135 suggests (Synolakis et al., 2007). The long wave typically associated with tsunamis implies a small error in the result and the analytical solution show excellent agreement in the experiment mentioned which implies that no large mass loss is present. A comment about mass conservation was added in the manuscript (p.9 line 11).

Q3. Regarding the water height positivity (h), is it guaranteed in the propagation stage even considering the cubic polynomial interpolation process?

A3. The propagation stage of the program is not used in the run-up calculation or on land areas. Coasts not flagged as "*Inundation*" have a wall boundary treatment. In the case of the run-up calculation, the surface gradient method is used. It is constructed as a monotone scheme with flux correction and a slope limiter to preserve height positivity (added in p.9 line 21). Dam-break problem tests with thin water layers have been satisfactory with the wave remaining positive even in shallow conditions. In addition, the new run-up benchmark problems (BP) added to the paper contribute as another demonstration of positivity preservation.

Q4. On other hand, what is the convergence order of the numerical models used on each area?

A4. The convergence of the method of characteristic used for propagation and the method for run-up are different. The propagation is 3rd order while the run-up is a 1.5 order. A measurement of this convergence for the parabolic bowl problem is included to illustrate this point (Fig. A). Figure included in the manuscript Fig. 3 (p.10).

[Figure]

**Fig A. Parabolic bowl problem cross section with $\epsilon = 10^{-4}$ on left panel. Water depth error for parabolic bowl problem on right panel**

Section 3

Q5. As the coordinate system used combines spherical coordinates and cartesian coordinates, how is treated the boundary between the considered domains on each coordinate system?

A5. The Cartesian coordinate system is used exclusively for the blocks set as "*Inundation*". These blocks are only found at level 7 (highest level) and only inside focal areas. While the focal areas can be user-defined in any size and located on any coast in the domain, by design the focal areas shall consist of just a few kilometers in length. In this way, the area represented by the inundation areas is extremely small compared to the total domain size. This makes it possible not to use a special boundary treatment between systems since the difference of the incoming wave is almost negligible.

Additionally, the domain used centered in the Indian Ocean does not extend to high latitudes that might introduce large discrepancies in the grid. In Fig. B, the current focal areas used by TRITON-G are shown with the approximate length in kilometers. As noted, they cover just a few kilometers, the largest case being Seychelles, which is just around 27 km by 17 km. We added a comment explaining this boundary in the revised manuscript (p12, line 3) as well as modified Fig. 5 (p.14).

[Figure]

**Fig. B. Focal Areas highlighted in green with the approximate length labeled in kilometers. Top Left: Sri Lanka. Top Right: Comoros. Bottom left: Mozambique. Bottom Right: Seychelles**

Q6.  Regarding the run-up calculation, in section 3.2 authors propose a technic based on considering (hu=hv=0) when h is less than a certain small fixed quantity. They confirm that the proposed implementation has been proven to be robust and stable under different benchmarks. Please, include a reference where this numerical technic is used.

A6. Some references (p. 9, line 20) about the thin layer technique and its implementation are:

- Vincent, Stéphane and Caltagirone, J.P and Bonneton, Philippe. (2001). "Numerical modeling of bore propagation and run-up on sloping beaches using a MacCormack TVD scheme." Journal of Hydraulic Research – 39: pp 41-49.
- Xin, L and Abdolmajid, M and Infants Sedano, J. "A Well-Balanced 2-D Model for Dam-Break Flow with Wetting and Drying" Journal of Fluid flow, Heat and Mass Transfer. Vol 1 (2014): pp: 30-37
- Wang, L.X. (1987) ''Composite-model of 1-d and 2d flows'', Advanced course and workshop on mathematical modeling of alluvial river, IRTCES-Beijing, China.

Q7. Finally, in this section it's included a parabolic bowl problem as validation problem. That's a good synthetic test but I consider that it's not enough to define a numerical model as validated when such numerical model that is going to be used with real topobathymetries in real cases, moreover when, as it's remarked by authors, the model is going to use in the RIMES context. In order to validate this numerical scheme to be used in real cases I suggest, for instance, the use of the inundation benchmark experiments proposed by the National Tsunami Hazard Mitigation Program (NTHMP) where problems with analytical solutions, laboratory experiments and real problems with field measurements are proposed. I would suggest studying the behavior of TRITONG at least in the "mandatory" benchmark problems: BP1, BP4, BP6, BP7 and BP9. Regarding the field measurement experiments, authors show in section 6.1.2 some inundation maps in different locations (Fig. 18 and 19) where the comparisons are made basically against other models. I think that it cannot be considered a numerical model as validated with field measurements by making comparisons basically with other models results. There are available many tsunami field measurements to validate the inundation process. In this sense you can consider cases where there are more available data than in the case studied in section 6.1.2. For instance: BP7, BP9 or many inundation scenarios related to the Tohoku, 2011 event where detailed data are available.

A7. We appreciate the suggestion to validate TRITON-G against the inundation benchmark problems. We have completed the results for benchmarks BP4, BP6, BP7 and BP9. We skipped benchmark problem BP1 considering that it is a one-dimensional analytical problem and we already showed results with good agreement for the parabolic bowl problem, which is a two-dimensional analytical problem. Additionally BP4 is the experimental version of BP1 and results for BP4 are now included. This serves to demonstrate the behavior obtained with TRITON-G under these conditions.

Regarding the map comparison with another model (Fig. 18 and Fig 19) we consider that is an acceptable way to show agreement since the results from these models are peer-reviewed and published (Supparsri et al., 2011). However, in order to produce a more complete validation of our model we agree with the importance of using tsunami field measurements and known benchmark

problems. For this reason, we also use BP7 and BP9 as comparison to validate the inundation process.

A new section was added to the manuscript to include results for BP9 (p.28, Section 6) while the rest of the benchmarks are located in the appendix (p. 44).

1.  **Benchmark problem #4: Solitary wave on a simple beach – Laboratory**

Figure C depicts the domain for this test. In this problem, the wave height H is located at a distance L from the beach toe. This test was replicated in a wave tank 31.73-cm-long, 60.96-cm-deep and 39.97-cm-wide at the California Institute of Technology. Several experiments with different water heights were performed. Benchmark Problem 4 (BP4) uses the datasets for $H/d = 0.0185$ non-breaking wave and $H/d = 0.30$ breaking wave for code validation. Results use dimensionless units with the help of parameters like length $d$, velocity scale $U = \sqrt{gd}$ and time scale $T = \sqrt{d/g}$.

[Figure]

**Fig. C Domain sketch for BP4 with slope 1:19.85 (figure taken from benchmark description)**

**1.1 Problem setup**

- *Parameters*: d = 1, g = 9.8, case A with $H/d = 0.0185$ and case C with $H/d = 0.30$.
- *Friction*: Manning coefficient set to 0.01
- *Computational domain*: the domain along $x$ direction spanned from x $= -20$ to x $= 80$
- *Boundary conditions*: the right side of the computational domain uses a non-reflective boundary condition.
- *Grid resolution*: the numerical results are solved with a resolution of $\Delta x = 0.1$
- CFL: 0.9
- *Initial condition*: the initial wave is computed based on the following equations for height (η) and velocity (u) respectively

$$\eta(x. 0) = Hsech^2[\gamma(x - x_s)/d] , \qquad (1)$$

$$u(x, 0) = -\eta(x, 0)\sqrt{\frac{g}{d}}. \qquad (2)$$

**1.2 Tasks to be performed**

To accomplish this problem, the following tasks should be performed:

1. Compare numerically calculated surface profiles at t/T=30:10:70 for the non-breaking case $H/d = 0.0185$ with the lab data (Case A).

2. Compare numerically calculated surface profiles at t/T=15:5:30 for the breaking case $H/d = 0.30$ with the lab data (Case C).

3. Compute maximum run-ups for at least one non-breaking and one breaking wave case.

**1.3 Numerical results**

We present the numerical results obtained using TRITON-G. Figure D shows the comparison between water surface level measured in the experiment and the modeled numerical results obtained by our model for times 30, 40, 50, 60 and 70 for case A ($H/d = 0.0185$). Our results show good agreement between the numerical simulation and the non-breaking experiment.

Table A shows the errors computed for the normalized root mean square deviation (*NRMSD*) and for the maximum wave amplitude error (*MAX*). The error values obtained by the NTHMP workshop models are also included for comparison (taken from Table 1-8, page 41 in (NTHMP, 2012) ). These values are divided into two columns, one with results for the non-dispersive models (*ND*) and the other with results for the non-dispersive and dispersive models together (labeled *ALL*). Errors obtained from our simulation tend to be similar or smaller than those errors obtained by other ND models, with just slight exception for time 70. Additionally, except for time 70 our errors are smaller than those obtained combining non-dispersive and dispersive mean error value.

Water level comparison for case C ($H/d = 0.30$) at times 15, 20, 25 and 30 is shown in Fig. E. Table B gathers the values for NRMSD and MAX errors for our numerical results and for the NTHMP workshop models. In this case, only the results of models that reported their errors are included (taken from Table 1-8, page 41 in (NTHMP, 2012) ).

For case C conditions, the shallow water equations are no longer appropriate for modeling and hydrostatic models tend to produce larger differences than non-hydrostatic ones. Our numerical results in general show good agreement with the experiment. The difference with the steepening of the crest that is noticeable in the results is expected from a hydrostatic model. In spite of that, this steeping in our model is not very large and it can trace the wave front well. Once the wave breaking occurs, our model can simulate reasonably well the run-up. This is also partly reflected in the small NRMSD error estimation obtained by our model after the wave breaking.

Maximum run-up for case A and case C were calculated. For the non-breaking case A, the obtained run-up value is 0.091 and for the breaking case C, the run-up estimated is 0.588. These values are plotted in Fig. F with a yellow and red dot respectively, it can be seen that both values lie well within the experimental results.

|  |  | NRMSD | | | MAX | | |
|---|---|---|---|---|---|---|---|
|  |  | TRITON-G | NTHMP | | TRITON-G | NTHMP | |
|  |  |  | ND | ALL |  | ND | ALL |
| H = 0.0185 |  |  |  |  |  |  |  |
| T | 30 | 8.8 | 11 | 11 | 4.0 | 6 | 4 |
| T | 40 | 6.6 | 9 | 8 | 4.8 | 3 | 3 |
| T | 50 | 3.5 | 6 | 5 | 7.4 | 13 | 7 |
| T | 60 | 3 | 4 | 5 | 1.4 | 1 | 3 |
| T | 70 | 11 | 33 | 16 | 13.5 | 15 | 9 |

**Table A. Model surface profile errors with respect to laboratory experiments for case A** $H/d = 0.0185$ **at times 30, 40, 50, 60, and 70. Results from the NTHMP workshop errors are divided into non-dispersive (ND) models and all models (ALL)**

|  |  | NRMSD | | MAX | |
|---|---|---|---|---|---|
|  |  | TRITON-G | NTHMP | TRITON-G | NTHMP |
| H = 0.3 |  |  | ALL |  | ALL |
| T | 15 | 11.3 | 7 | 5.4 | 6 |
| T | 20 | 5.9 | 9 | 23.3 | 11 |
| T | 25 | 6.5 | 6 | 11.1 | 10 |
| T | 30 | 2.9 | 4 | 1.4 | 6 |

**Table B. Modeled surface profile errors with respect to laboratory experiments for case C** $H/d = 0.30$ **at times 15, 20, 25 and 30. Results from the NTHMP workshop model errors available are shown (ALL)**

[Figure]

**Fig. D. Comparison of numerically calculated free surface profile at different dimensionless times for the non-breaking case $H/d = 0.0185$**

[Figure]

**Fig. E. Comparison of numerically calculated free surface profile at different dimensionless times for the breaking case $H/d = 0.30$**

[Figure]

**Fig. F. Scatter plot of non-dimensional maximum run-up from a total of more than 40 experiments conducted by Y. Joseph Zhan** (Synolakis, 1987)**. Orange point indicates TRITON-G result for the breaking case $H/d = 0.30$ and yellow point indicates the result for the non-breaking run-up case $H/d = 0.0185$**

**2. Benchmark problem #6: Solitary wave on a conical island – Laboratory**

The goal of this benchmark is to compare computed model results with laboratory measurements obtained during a physical modeling experiment conducted at the Coastal and Hydraulic Laboratory Engineer Research and Development Center of the U.S. Army Corps of Engineers. The laboratory physical model was constructed as an idealized representation of Babi Island, in the Flores Sea, Indonesia, to compare with Babi Island run-up measured shortly after the 12 December 1992 Flores Island tsunami (Yeh et al., 1994). Figure G shows schematics of the experiment.

**2.1 Tasks to be performed**

To accomplish this benchmark, it is suggested that, for

      Case A: water depth d= 32.0 cm, target H=0.05, measured H=0.045

      Case B: water depth d= 32.0 cm, target H=0.20, measured H=0.096

      Case C: water depth d= 32.0 cm, target H=0.05, measured H=0.181

model simulations be conducted to address the following:

1. Demonstrate that two wave fronts split in front of the island and collide behind it

2. Compare computed water levels with laboratory data at gauge 6, 9, 16 and 22

3. Compare computed island run-up with laboratory gauge data

**2.2 Problem setup**

- *Computational domain (in meters)*: [-5,23] × [0, 28]
- *Boundary condition*: open boundaries
- *Initial condition*: same solitary wave as proposed in BP4 with the correction for two dimensions
- *Grid resolution*: the numerical results presented are solved with a resolution of $\Delta x = 0.05$
- *CFL*: 0.9
- *Friction*: Manning coefficient set to 0.02

[Figure]

**Fig. G Basin geometry and coordinate system. Solid lines represent approximate basin and wave maker surfaces. Circles along walls and dashed lines represent wave-absorbing material**

[Figure]

**Fig. H Snapshots at several times showing the wave front splitting in front of the island and colliding behind it for BP6 case B.**

**2.3 Numerical results**

We present the numerical results obtained using TRITON-G for the three cases (A, B and C) except for the splitting-colliding item. For this item, Figure H shows the wave front splitting in front of the island and then colliding again behind it for case B (H=0.096), analogue behavior was obtained for the other two cases.

| | | NRMSD | | | MAX | | |
|---|---|---|---|---|---|---|---|
| | | TRITON-G | NTHMP | | TRITON-G | NTHMP | |
| Case A | | | ND | ALL | | ND | ALL |
| Gauge | 6 | 10 | 6 | 7 | 4 | 9 | 8 |
| Gauge | 9 | 9 | 7 | 8 | 4 | 14 | 10 |
| Gauge | 16 | 7 | 10 | 9 | 5 | 10 | 12 |
| Gauge | 22 | 9 | 8 | 8 | 4 | 25 | 18 |
| | | | | | | | |
| Case B | | | | | | | |
| Gauge | 6 | 10 | 8 | 8 | 7 | 6 | 6 |
| Gauge | 9 | 9 | 8 | 8 | 2 | 7 | 9 |
| Gauge | 16 | 9 | 7 | 7 | 14 | 7 | 7 |
| Gauge | 22 | 8 | 9 | 9 | 6 | 40 | 27 |
| | | | | | | | |
| Case C | | | | | | | |
| Gauge | 6 | 13 | 10 | 8 | 3 | 6 | 5 |
| Gauge | 9 | 12 | 11 | 11 | 2 | 9 | 13 |
| Gauge | 16 | 10 | 9 | 8 | 4 | 3 | 3 |
| Gauge | 22 | 9 | 8 | 8 | 10 | 18 | 15 |

**Table C. Water level time series TRITON-G model errors with respect to laboratory experiment data for case A, B and C. Mean values obtained for the performing NTHMP models is separated in non-dispersive models (ND) and dispersive and non-dispersive models together (ALL)**

Water level comparison uses values for gauges 6, 9, 16 and 22 for each of the 3 cases. Gauge 6 is located at $(9.36, 13.80, 31.7)$, Gauge 9 is located at $(10.36, 13.80, 8.2)$, Gauge 16 is located at $(12.96, 11.22, 7.9)$ and Gauge 22 is located at $(15.56, 13.80, 8.3)$.

Numerical results for Case A, B and C are shown in Figures I, J and K respectively. In the three cases results were stable and in good agreement with the experimental values. The incident wave height and arrival time was captured well for all gauges. Similarly as with BP4, the steepening of the wave with increasing H is expected in a non-hydrodynamic model. After the wave hit the island, some differences between experimental and model wave are noticeable as the initial wave height increased. These oscillations in the experimental data represent the effects of dispersion, which our non-dispersive numerical method is not designed to capture. Despite this, the modeled waves show good agreement with the shape of the experimental waves and the errors estimated tend to be small.

[Figure]

**Fig. I Comparison between computed and measured water levels at gauges 6, 9, 16 and 22 for case A (H=0.045)**

[Figure]

**Fig. J Comparison between computed and measured water levels at gauges 6, 9, 16 and 22 for case B (H=0.096)**

[Figure]

**Fig. K Comparison between computed and measured water levels at gauges 6, 9, 16 and 22 for case C (H=0.181)**

[Figure]

**Fig. L Comparison between computed and measured run-up around the island for the three cases in BP6**

|  | NRMSD | | | MAX | | |
|---|---|---|---|---|---|---|
|  | TRITON-G | NTHMP | | TRITON-G | NTHMP | |
| Run-up | | ND | ALL | | ND | ALL |
| Case   A | 9 | 18 | 18 | 0.6 | 12 | 7 |
| Case   B | 19 | 21 | 18 | 9 | 2 | 5 |
| Case   C | 20 | 12 | 11 | 14 | 5 | 5 |

**Table D. Run-up TRITON-G model errors with respect to laboratory experiment data for case A, B and C. Mean error values obtained by the performing NTHMP models are separated in non-dispersive models (ND) and all models (ALL)**

Table C gathers the normalize root mean square deviation (*NRMSD*) error and the maximum wave height (*MAX*) error. For comparison, mean errors obtained by the participating models in the NTHMP workshop are also included. These are separated in two columns, one for non-dispersive (ND) models and the other for non-disperse and disperse models together (ALL).

NRMSD errors for our model tend to be not very large and in similar range than those of the other non-dispersive models. In the case of the maximum height error (MAX), in almost all cases our model produced smaller error values than the non-dispersive model counterparts. Additionally, in most cases our MAX errors are smaller than those errors of the combined non-dispersive and dispersive mean values.

Figure L shows the comparisons between computed and experimental run-up around the island for the three cases. Case A represent the best agreement with the experimental values. Differences increased with steeper wave cases B and C as several reflections and refraction possibly occur in the basin.

Table D gathers the errors obtained by our model and by the participating models in the NTHMP workshop for run-up cases A, B and C. Figure L showed the good agreement for Case A and this is also reflected in the NRMSD and MAX error results. Both values are considerably smaller than those errors obtained by the NTHMP non-dispersive (ND) models and by the non-dispersive and dispersive together (ALL). For cases B and C, the errors tend to be larger but still similar to those obtained by other non-dispersive models. In all cases, the error stayed below the 20% recommended criteria.

**3 Benchmark problem #7: The tsunami run-up onto a complex 3-D beach.  Laboratory.**

A laboratory experiment using a large-scale tank at the central Research Institute for Electric Power Industry in Abiko, Japan was focused on modelling the run-up of a long wave on a complex beach near the village of Monai (Liu et al., 2008). The beach in the tank was a 1:400-scale model of the bathymetry and topography around a very narrow gully, where extreme run-up was measured.

**3.1 Problem setup**

The following parameters were used for the computation:

- *Grid resolution*: 393×244 was used with the same resolution 0.014 m as the bathymetry.

- *CFL*: 0.9

- *Initial condition*: water at rest.

- *Friction*: Manning coefficient set to 0.01

- *Boundary conditions*: Solid wall boundary used at the top and bottom. At the left boundary, the given initial wave (shown in Fig. M) was used to specify the condition up to time t=22.5 s, after that it became a wall boundary condition.

[Figure]

**Fig. M Prescribed input wave for the left boundary condition defined from t=0 to t=22.5 s**

**3.2 Tasks to be performed**

To accomplish this benchmark it is suggested to:

1. Model propagation of the incident and reflective wave accordingly to the benchmark-specified boundary condition.

2. Compare the numerical and laboratory-measured water level dynamics at gauges 5, 7 and 9.

3. Show snapshots of the numerically computed water level at the time synchronous with those of the video frames.

4. Compute maximum run-up in the narrow valley.

**3.3 Numerical results**

This section presents the numerical results for BP7 obtained with TRITON-G to achieve the required tasks.

[Figure]

**Fig. N Water level comparison for BP7 between experiment and our model for gauges 5, 7 and 9**

The comparison with the three requested gauges 5, 7 and 9 is shown in Fig. N from $t = 0$ to $t = 25$ s. For the three cases, good agreement is found between modeled and experimental wave.

Values for the normalized Root Mean Square deviation error (NRMSD) and maximum wave amplitude error (MAX) were estimated for the gauge comparison. For gauge 5, the NRMSD error is 10% and MAX is 0.89%. For gauge 7, NRMSD is 10% and MAX is 4.81%. For gauge 9, the NRMSD error is 6.57% and MAX is 2.66%.

[Figure]

**Fig. O Comparison between extracted movie frames (left) and TRITON-G simulation (right) for times 15, 15.5, 16, 16.5 and 17 seconds**

Comparison with the extracted movie frames is shown in Fig. O. In the left column are the five frames provided from the laboratory recording. These are frames 10, 25, 40, 55 and 70, extracted from the video with a 0.5 s interval. We found good agreement in time and space for times 15 s to 17 s in 0.5 s increments, shown in the right column. The side-by-side comparison shows that the modeled wave follows the experimental wave front well. Additionally, the model captures the rapid run-up/run-down in the narrow gully.

Finally, the data provided by the benchmark workshop include a series of experiment tests for maximum run-up. Its maximum run-up is recorded at $\times= 5.1575$ and $y = 1.88$ m with an average value of approximately 0.09 m. In comparison, our numerical result recorded a maximum run-up at around $t = 16.5$ with a height of 0.0936 m at $\times= 5.15$ and $y = 1.88$ m.

**4 Benchmark Problem #9: Okushiri Island Tsunami - Field**

This benchmark problem (BP9) is based on the data collected from the Mw 7.8 Hokkaido-Nansei-Oki tsunami around Okushiri Island in Japan in 1993. The goal is to compare computed model results with the field measurements.

**4.1 Problem setup**

The following parameters were used for the computation:
- *Bathymetry*: taken from databases provided by (NTHMP, 2012), interpolated where necessary.
- *CFL*: 0.9
- *Simulated time*: 60 minutes
- *Initial condition*: source generated from the database provided by DCRC (Disaster Control Research Center) Japan solution DCRC17a, described in (Takahashi, 1996).
- *Boundary conditions*: open boundaries at the four domain edges.
- *Friction*: Manning coefficient set to 0.02
- *Computational domain*: a mesh refinement is used on the entire domain (shown in Fig. P). Seven levels are used in total. The resolution of base level 1 is 450 m and the resolution of level 7 is approximately 7 m. Dry blocks that did not take part in the computation were removed in the mesh generation process.

[Figure]

**Fig. P Left: entire domain refined mesh containing seven levels. Right: zoom on Okushiri Island. Higher resolution used around Monai Valley at level 7 (7 m approx.) and Aonae region at level 6 (14 m approx.)**

**4.2 Tasks to be performed**

This benchmark requires the following tasks to be performed:

1. Compute run-up around Aonae

2. Compute arrival of the first wave to Aonae

3. Show two waves at Aonae approximately 10 minutes apart; the first wave came from the wet, the second wave came from the east

4. Compute water level at Iwanai and Esashi tide gauges

5. Maximum modeled run-up distribution around Okushiri island

6. Modeled run-up height at Hamatsumae

7. Modeled run-up height at a valley north of Monai.

**4.3 Numerical results**

In this section, we present the numerical results obtained with TRITON-G for benchmark problem #9.

**4.3.1 Run-up around Aonae**

The maximum inundation around Aonae peninsula modeled during the simulation is shown in Fig Q. Contours every 4 meters are drawn to show the outline of the topography. Maximum

inundation height computed was nearly 15 meters but the scale used is set to the upper limit of 10 m to highlight the areas where major inundation occurred.

The west side of the peninsula received the impact of the first wave, which produced the largest inundation height. Maximum values of nearly 15 m were obtained in the simulation. Despite a relatively lower inundation height in the east side of the peninsula, deep penetration was found due to the flatter topography in this area. The inundation on the east side was mainly produced by the second wave coming from the east. The south side of the peninsula experienced the impact of both first and second waves and run-up of over 12 m was estimated.

[Figure]

**Fig. Q Inundation map of Aonae region with 4-m contours of bathymetry and topography**

**4.3.2 Arrival of first wave to Aonae**

The arrival of the first wave at Aonae peninsula is shown in Fig. R. This wave is coming from the west. Snapshots are approximately 5 seconds apart at times 4.9 min and 5.0 min to illustrate the wave arrival. From these snapshots, we estimate that the wave made impact at around 5 minutes after the tsunami generation.

[Figure]

**Fig. R Arrival wave at Aonae peninsula coming from the west, snapshots of the wave at times 4.9 min and 5.0 min after tsunami generation**

**4.3.3 Two waves arriving at Aonae**

[Figure]

**Fig. S Two waves arriving at Aonae peninsula. Left: first wave coming from the west arrived at around t=5 min. Right: second wave coming from the east arrived at around t=16 min**

The two waves arriving at Aonae peninsula are shown in Fig. S. The first one came from the west (Fig. S left) and made impact at around 5.0 min after the tsunami generation. The second major wave to hit the peninsula came from the east and made impact at around 16 min (Fig. S right). Slightly over 10 minutes separated the first and second wave.

**4.3.4 Tide gauge comparison at Iwanai and Esashi**

Comparison between computed and observed water levels at Iwanai and Esashi tide gauges is presented in Fig. T. The arrival time of the computed wave shows good agreement for Esashi station. The computed wave positive and negative phases also follows rather well the observed values. In the

case of Iwanai station the arrival time is slightly sooner than the observed however the observed wave phase is followed generally well in the computed results. The discrepancies between observed and computed values can be attributed to several reasons. Inaccuracies in the source used for the initial condition can influence greatly the result. Additionally, lack of realistic bathymetry including man-made structures around the area can affect the results as well.

[Figure]

**Fig. T Water level comparison between observations and TRITON-G results for Esashi (upper panel) and Iwanai (lower panel) tide gauges.**

Inserted in each panel of Fig. T are the estimated errors for the gauge comparison. The maximum wave amplitude error for Esashi station is 16.27% and for Iwanai 3.19%. These are considerably lower than the mean values obtained by the models reported in the workshop (NTHMP, 2012) of 43% and 36% respectively. Although no values are reported in (NTHMP, 2012), the NRMSD error is also estimated for our model and included in the panels, both values are under 20%.

**4.3.5 Maximum run-up around Okushiri**

The computed maximum run-up distribution around Okushiri Island is shown in Fig. U. Observations were taken from (Kato and Tsuji, 1994). Good agreement is found between observed and computed values around the coast. Most values are within the observed range or within a small difference from the field measurement. The simulation seems to capture well the variations that occurred along the coast.

[Figure]

**Fig. U Computed and observed run-up values in meters along the Coast of Okushiri Island.**

The model could simulate well the maximum run-up observed around Monai valley within a reasonable 15% error. The major differences are found in the southwest side of the island where run-up values were underestimated with larger difference. The discrepancies could be explained by the use of different grid around the island coast. Additionally, the lack of an accurate high-resolution bathymetry database everywhere can also influence the computed values as well as an inaccurate initial condition.

**4.3.6 Run-up height at Hamatsumae**

The maximum inundation map for Hamatsumae region is shown in Fig. V. Topography and bathymetry contours are outlined every 4 meters. A grid resolution of approximately 14 m was used for this region. Near the center of the region and to the east, run-ups of nearly 16 meters were computed. Additionally, inundation values ranging from 8 to 10 meters were obtained which match well with field observations.

[Figure]

**Fig. V Inundation map of Hamatsumae region with 4-m contours of bathymetry and topography.**

**4.3.7 Run-up height at a valley north of Monai**

[Figure]

**Fig. W Inundation map for the valley north of Monai with 4-m contours of bathymetry and topography.**

The maximum inundation map for the valley north of Monai is shown in Fig. W. Topography and bathymetry contours are outlined every 4 meters. A grid resolution of approximately 7 m was used for this region. Inundation of around 26 m was computed, relatively close to the 30.6 m observed in the field data.

Q8. Regarding the generation process described in section 3.3, authors use the coseismic deformation proposed by Smylie and Manshiha, 1971. I'm surprised at this point because in [Okada, Y. (1985), Surface deformation due to shear and tensile faults in a half-space, Bulletin of the Seismological Society of America, 75(4), 1135-1154] this work is extended, and it's provided also an analytical

way to get the "Okada model" coseismic deformation that can easily be extended to CUDA in the GPU context. I would like to know the reason for not considering Okada as model for the generation process.

A8. Perhaps presenting our model as a *three-step simulation* program might have been misleading. The goal is our model focus on propagation and inundation. Porting to CUDA does not constitute any constraint however the fault generation was beyond our decision. The fault generation and theory used is based on RIMES internal decision. Their fault parameters are submitted to our model directly in order to start the simulation. We modified the manuscript to make this more clear (p.11, line 8). In future work we will focus on studying and including different fault theories such as the Okada model or dynamic generation, ported to GPU.

Section 4

Q9. In section 4, authors use a tree-based mesh refinement similar to the AMR technic used by R.J. LeVeque in GeoClaw. In this case the refinement is not made automatically as in GeoClaw, but it's customized to be refined in coastal areas and focal areas. It's an interesting alternative to the use of nested meshes when more detailed information is required in certain coastal areas.

In p14, line 9-10 it is discussed the number of blocks necessary under the considered resolution and according the 65 x 65 node-centered cells if the number of blocks is 230,000 you would have 971,750,000 node-centered cells. How many information is stored on each node-centered cell (in double precision) to represent over 100GB of memory space?

A9. For that test case, each cell stores approximately 112 bytes in double precision. This includes 14 different values to store information about latitude, longitude, the governing variables ($h^t$, $hu^t$, $hv^t$), the next time-step variables ($h^{t+1}$, $hu^{t+1}$, $hv^{t+1}$), the constant $H$, the bathymetry ($z$), manning coefficient and three constant values used for optimization.

Later, the introduction of focal areas reduced the domain mesh size and increased the available memory. Using this freed memory, three more constants were stored per cell. Additionally, the inundation output blocks were stored in GPU memory during the simulation for optimization.

Section 5

Q10. I have some comments, but my main concern is that I think that this section is out of the scope of this journal and my recommendation would be to publish it in a journal more related to this field. Anyway, I will make some comments regarding this section. To my knowledge, the overall GPU

implementation has been solved in a very efficient way, particularly the implementation of the pipe asynchronous output that is crucial to deal with the GPU-CPU traditional bottleneck.

A10. We appreciate your comment and suggestion however, while we understand that the GPU computing is not the scope of this journal we think that including the GPU implementation is very important since is a key element of our research. Tsunami forecasting requires a fast result and there lies the relevance of implementing our GPU calculation. For this reason, we give a brief description about GPU computing and a general explanation of our CUDA implementation and optimizations.

Q11. In p23, line 8 it is showed that you use CFL 0.8, is not stable the implemented model for CLF nearer to 1? On other way, in p24, line 5 it is used $\Delta t = 1.6$ for blocks with levels over 3. Is this consistent with the CFL condition? Or, can you ensure the stability of the numerical scheme under this assumption?

A11. To answer the first part of the question, in general it is stable to use a CFL condition closer to 1. The Semi-Lagrangian scheme used for the propagation stage allows a large time-step. Additionally, the added inundation benchmark problems (see Question 7) use CFL = 0.9, producing good results while being stable.

CFL values closer to 1 tend to produce stable results if the bathymetry varies smoothly. Based on our experience, some instabilities may arise in cases where the bathymetry presents sudden large gradients or very irregular shapes. The Indian Ocean bathymetry used in our research contain several of these cases. A common solution is to smooth the bathymetry before usage. Clearly, this introduces changes in the measurements and simulation. However, in order to keep the results as realistic as possible, we decided not to smooth the bathymetry and instead trade off a higher CFL value.

About the second part of the question, the CFL condition is consistent when using $\Delta t = 1.6$ and numerical stability is ensured in this situation. In order to illustrate this, Table E contains four columns with the values used for the simulation. The second column shows the maximum $\Delta t$ allowed in each level using CFL = 0.8. In order to speed up the computation, a sub-cycling method was introduced. A global $\Delta t = 1.6$ s is set to calculate the number of cycles that blocks in each level must take. The value of 1.6 is chosen to avoid a sub-cycling overhead since around 80% of the blocks are distributed in levels 1 to 4. The third column in Table E shows the resulting number of sub-cycles per level (*ns*). Finally, the fourth column shows the new CFL values obtained for each level when using sub-cycling (*S.C. CFL*). As it can be seen, in all levels, the CFL condition is less than 1 and stability is ensured.

We included Table E in the manuscript (p. 25, line 10) with an explanation to make clearer that stability is ensured when using sub-cycling.

| Level | Max Δt (CFL=0.8) | ns (Δt =1.6) | S.C. CFL |
|---|---|---|---|
| L1 | 10.71 | 1 | 0.12 |
| L2 | 5.13 | 1 | 0.25 |
| L3 | 2.37 | 1 | 0.54 |
| L4 | 1.65 | 1 | 0.78 |
| L5 | 0.95 | 2 | 0.68 |
| L6 | 0.55 | 4 | 0.59 |
| L7 | 0.26 | 8 | 0.39 |

**Table E. CFL values used after introducing sub-cycling (S.C. CFL) for each of the seven levels. The second column shows the maximum Δt per level using CFL = 0.8 and the third column shows the number of sub-cycles (ns) required in each level when using Δt =1.6**

Q12. In section 5.3.2 it is showed the runtime performance by comparing results obtained with two different GPU architectures: Tesla K80 and Tesla P100. It must be remarked that the configuration of the computation nodes that you are using is different for each architecture. While the Tesla P100 nodes have 4 GPU's per node, the Tesla K80 nodes have 2 GPU's per node. Unless your tests are using only two P100 per node, the network communications will reduce the performance between the K80 in front of P100. What kind of network is being used in these clusters?

A12. In the case of *Tsubame 3.0,* there are four Tesla P100 GPUs per node and the network is Intel Omni-Path HFI 100Gbps. In the case of the K80 machine used, there are four cards in one node (eight GPU in total), connected through PCI-Express 3.0. Confusion might arise from the term "Tesla K80 (12GB×2)". The values in parenthesis refer to the memory distribution. One K80 card includes two GPU chips, each with 12 GB. We modified the manuscript to describe this machine specification more clearly (p.26, lines 12 and 13).

We agree that the performance might be different depending on the GPU/node distribution. The aim on comparing between these machines is to demonstrate the portability of our code from an older architecture to a much newer one. Additionally, it serves to show a more general performance results by using more than one more machine and to highlight that the code can be implemented on different hardware without requiring changes or creating any problem (p.26, line 15).

Q13. Another point is, are output data being stored into hard disks in these tests or are they related only to pure computation time?

A13. All runtime measurements include the output time storing into hard disks. We included a mention to this fact in the revised manuscript (p.27, line 7).

Section 6

Q14. Finally, my main comments about section 6 have been included in my recommendations for section 3.

A14. Once again, we thank the reviewer for their important recommendation. We followed the advice given about section 3 and the full answer including the results of the new benchmark problems can be consulted in Question #7 (Q7).

Q15. Anyway, I have some specific comments about this section. It would be nice if the color scale used in Fig. 18 and 19 are the same for each subplot. With these graphics we can compare the inundation extensions, but it's difficult to compare the inundation height when different scales are used on each graphic. On other hand, and as I pointed before, this test is not enough to say that the model is validated, so I don't agree with the sentence of p33, lines 8-9.

A15. The subplots in Fig. 18 and 19 have been redone to match better the scales for comparison. The new subplots are shown in Fig. X and Fig. Y respectively. The new figures are included in the revised manuscript (p.42, Figure 29).

We have modified the sentenced in p33 about the inundation validation using just one test. It is now noted (p.43, line 19) that the additional standard inundation benchmark problems computed produced good results and served as complementary demonstration of TRITON-G's ability to estimate tsunami inundation. A mention to the new benchmark results are also included in the conclusion as support for the validation (p.44, line 7).

[Figure]

[Figure]

**Fig X. Inundation comparison for Hambantota, Sri Lanka. Top: RIMES model. Bottom: TRITON-G model.**

[Figure]

[Figure]

**Fig Y. Kamala (North) and Patong (South) inundation maps comparison. Top: inundation result by** (Supparsri et al., 2011)**. Bottom: TRITON-G inundation result.**

**Corollary**

Author's comment: Additional to all the reviewers' suggestions, we decided to remove the paragraph about the circular shoal benchmark from the original manuscript in page 26, from line 11 to 17. With the introduction of several new benchmark problems (Reviewer #1 Question #7) and the modifications to the original manuscript, it felt unnecessary to keep this reference since the new results covered far more than what this benchmark offered.

**Response to Reviewer #2**

We want to thank the reviewer for using their precious time to check our paper and for giving useful comments to improve our work. Please find below our replies to your comments.

General Comments

Q1. The paper presents an operational model for the fast simulation of the generation, propagation and inundation of tsunamis in wide areas by exploiting modern multiGPU hardware. The model is tested and compared under a real tsunami scenario, obtaining a nice performance results from the operational point of view. The implementation of this operational model on a cluster of multiGPU computers involves the suitable integration of numerical schemes (MOC with dimensional splitting to solve spherical SWE for the Tsunami propagation, Surface Gradient Method to solve the cartesian SWE for inundation areas, ...) and computing techniques (quadtree-based mesh refinement to save resources, Hilbert Space-filling curves to preserve locality in the parallel partitioning, CUDA for GPU programming and MPI for remote communication, overlapping the computation in GPU and the generation of output files and rendering in CPU, etc.) to obtain an efficient complete CPU-multi-GPU operational model for Tsunami forecasting. This model would make it possible a very fast simulation which can help in the early identification of the tsunami consequences. In my opinion, the techniques which are presented and the scientific data which are included are coherent and relevant and can be useful to scientists working in this area because all the approaches and techniques are devised in conjunction to perform very quickly realistic simulations. Although the paper is well organized and written, the reading of several pieces of the sections which explain the multiGPU implementation is not easy to understand and several implementation decisions which are presented are not clear. Moreover, in Section 5, I think that the use of technical and English language should be checked (several corrections are included in the Section of Technical Corrections).

A1. We appreciate your kind description of our work. We considered your suggestion to check the use of technical English in section 5 (p.17) and made the section about multi-GPU implementation more clear where possible. We appreciate your technical corrections in this matter as well.

It should be noted that even though one of the key elements of our work is GPU computing, the scope of the journal is not this area. For this reason, we tried to find a balance giving an appropriate description of our implementation without getting into extensive details. We tried to focus on the model and on the simulation results.

 On the other hand, to intend the validation of the operational model with a real tsunami scenario when the input data are not sufficiently accurate is very ambitious.

A2. In order to have more validation data we have now included two sections in the manuscript to show results of several standard benchmark problems. Not having accurate enough initial input data is always an issue for all tsunami simulation models. Currently, the best approach to validate models consists on comparing results with existing analytical solutions and experimental data. For this reason we follow the NOAA Technical Memorandum OAR PMEL-135 (Synolakis et al., 2007) where several standard benchmark problems (BP) are given. We also thank Reviewer #1 for this suggestion.

Section 3 presents the results for the analytical 2D Parabolic Bowl benchmark. A new section was added to the manuscript to include results for benchmark problem 9 (p.28, Section 6) while the rest of the benchmarks are located in the appendix (p.44). The benchmark problems added are:

- BP4, Solitary wave on a simple beach
- BP6, Solitary wave on a conical island
- BP7, The tsunami run-up onto a complex 3D beach
- BP9, Okushiri Island Tsunami

**Specific Comments**

Q3. In Section 1, it would be interesting to include a comparison with previous works related with the multiGPU simulation of tsunamis to obtain faster-than-real-time results.

A3. We appreciate the suggestion; however, we consider that this kind of comparison would be unfair to do. For instance, each model utilizes different numerical schemes; the machines used might have different specifications; the domain used in each case might be different and in our case, we used an AMR-like technique for mesh refinement when grid nesting is more common.

However, we have included as references in section 1 (p.2, at lines 8,13, 16 and 27) other works that use GPU in their simulations. Two examples are:

- Vazhenin, A., Lavrentiev, M., Romanenko, A. and Marchuk, A.: Acceleration of tsunami wave propagation modeling based on re-engineering of computational components, International Journal of Computer Science and Network Security, 13, 32–70, 2013.
- Macías, J., Castro, M. J., Ortega, S., Escalante, C. and González-Vida, J. M.: Performance benchmarking of Tsunami-HySEA model for NTHMP's inundation mapping activitie, Pure and Applied Geophysics, 174, 3147–3183, 2017.

Additionally, we have included new benchmark results in the manuscript (Section 6 and Appendix) that are considered standard in the tsunami field. Using the data presented in the National Tsunami Hazard Mitigation workshop (NTHMP, 2012), error comparison with results of other models was included in our discussion and figures when available.

Q4. In Section 5.1., the description of the configuration of the main CUDA kernels (second paragraph of the section 5.1.) is not easy to understand. A graphical description of the configuration and a description of the calculations assigned to each CUDA thread (relating this section with section 3 and 4) would be very useful to understand it.

A4. A graphical description of the CUDA kernel, shown in Fig. A, has been added to the revised manuscript (p.18, Fig. 7). A more clear explanation was included as well (p.18, lines 9-16).

The grid is composed of 16 CUDA blocks in *y*-direction, each with 4 threads for 64 threads in total. In *x*-direction, the grid has one CUDA block with 64 threads.

One CUDA block processes a portion equal in size of the mesh block. One CUDA thread computes one mesh block cell. The specific calculation varies depending on the block type (*Wet*, *Dry*); however, the configuration remains the same. In both cases, threads compute the governing equations described in section 3.1. The main difference occurs in the case of a *Dry* block; in this case, cells that represent land or coastline compute a reflective wall boundary.

[Figure]

**Fig. A Mesh block computation using CUDA kernels. Each CUDA block is made of 64×4 threads and computes a portion of the mesh block. One CUDA thread computes one mesh block cell**

Q5. The specialized kernel types presented in section 5.1.2. could affect the load balancing between GPUs because the computational execution cost of each kernel type on a mesh block would be possibly different. I do not know if this fact is taken into account for the considerations included in section 5.2.

A5. Thank you for pointing this detail. This fact is taken into account during the load balancing, by assigning a different weight to the space-filling curve (SFC) based on the block type. This was not mentioned explicitly in the manuscript. A mention to this has been added in section 5.1.3 where SFC weights are discussed with sub-cycling (p.25, line 24).

Q6. I think it would be interesting to report graphically execution times for each particular GPU in order to evaluate the effectiveness of the domain partitioning and even to rethink the approach by designing a dynamic load balancer.

A6. Without being a detailed dynamic load balancer, our model includes this feature. During the mesh generation, blocks are assigned a different weight based on its type and the sub-cycling number. This weight is used in the space-filling curve to find a good domain partition. Not being a static process, this means that if a new domain mesh is required, the program will balance the new load.

[Figure]

**Fig. B GPU execution time with and without load balance**

Evidently, the load balance is problem dependent. However, we include a chart show in Fig. B with the balancing results for our Indian Ocean domain using 4 focal areas (p.26, Fig 15). The effect of including the load balance can be seen on the right side of the chart. All GPUs spend almost the

same time to execute a time-step. This avoids large overheads created by one GPU idling waiting for another to complete the tasks (p.26, line 1).

Q7. In Section 5.3.2., the configuration for the network which interconnects the TeslaP100- based nodes and the Tesla K80 nodes should be included to analyze Fig. 15.

A7. In the case of *Tsubame 3.0,* there are four Tesla P100 GPUs per node and the network is Intel Omni-Path HFI 100Gbps. In the case of the K80 machine used, there are four cards in one node (eight GPU in total), connected through PCI-Express 3.0. These network configurations have been added to the manuscript (p.26, line 12 and 13).

Q8. In Section 5.3.2. and in the Conclusions, authors underline evidences about the wall clock time and the speedup which are obtained with 3 GPUs. However, the particular performance results for 3 GPUs are not reported and they are not included in Figure 15.

A8. Thank you for noticing this detail. The runtime for 3 GPUs with K80 cards is 39.96 min and 12.1 min with P100 cards. These values have been now reported in the manuscript in section 5.3.2 (p.28, Fig 17 and p.27, line 15). Additionally, they have been included in Figure 15; the modified figure can be seen in Fig. C.

[Figure]

**Fig. C Wall clock comparison of 10-hour simulation on Tesla K80 and Tesla P100**

Q9. In Section 5.3.2., absolute performance measures on one GPU for the main kernels (it can be obtained by using the Nvidia CUDA profiler) could be useful to evaluate the efficiency of the CUDA implementation.

A9. We measured the FLOP/s performance of the main kernels for one GPU. The results obtained are shown in Table A, where *Inund* stands for Inundation kernel, *Wall* stands for the wall kernel, *Wet* for the *Wet* kernel and *X* and *Y* for the direction of the computation equivalent to longitude and latitude respectively. These results have been added to the revised manuscript (p.27, line 1 and p.27 Table 2).

| Kernel | GFLOP/S |
|--------|---------|
| WallX  | 549.57  |
| WallY  | 549.56  |
| WetX   | 706.98  |
| WetY   | 712.51  |
| Inund  | 87.12   |

**Table A. Kernel performance for one GPU in Giga FLOP per second**

Q10. In section 6.1.2., the figure 18 Top which presents the RIMES results for Hambantota Inundation is not introduced textually in page 31, line 18. On the other hand, it is difficult to compare visually the RIMES and TRITON G inundation maps if the colours are used in a completely different manner for each map.

A10.

[Figure]

[Figure]

**Fig. D Inundation comparison for Hambantota, Sri Lanka. Top: RIMES model. Bottom: TRITON-G model.**

We reordered and modified the paragraph in page 31 to introduce textually RIMES' figure (p.41, line 7).

Additionally, we modified TRITON-G figure's color scale to make it match better with RIMES' scale (p.41, Fig 28). The result is shown in Fig D in these notes.

Q11. - Page 13, Section 4.1, line 7: "blocks close to the coast until reaching a target ..."

A11. Changed sentence accordingly (p.13, line 17).

Q12. - Page 15, Line 7,8: The sentence is not clear.

A12. Sentences were rewritten to make them more clear (p. 15, line 17).

"Additionally, all dry blocks at Level 7 (highest resolution) that are inside a FA are considered inundation areas. This implies that run-up is computed on the coastlines instead of using a reflective boundary."

Q13. - Page 17, Line 20; The sentence is not easy to understand.

A13. Sentence rewritten (p.17, line 28).

 "CUDA provides kernels as the way to define functions that are executed in parallel on GPU."

Q14. - Page 17, Line 21: "organized in a grid of blocks of CUDA threads ..."

A14. Changed sentence accordingly (p. 17, line 28).

Q15. - Page 18, Line 8: "branch diversion" is not the usual term. "branch divergence" is more frequent in this context.

A15. Thank you for the suggestion. Changed sentence accordingly (p.19, line 8).

Q16. - Page 18, Line 12: A comma in the sentence after "speed up" would help in order for it to make sense.

A16. Changed sentence accordingly (p. 19, line 12).

Q17. - Page 18, Line 18: "... This way the kernels can be launched ...".

A17. Changed sentence accordingly (p.19, line 18).

Q18. - Page 19, Line 5: "... is illustrated in Fig. 7 ...".

A18. Changed sentence accordingly (p.20, line 5).

Q19. - Page 23, Line 5: "... was introduced in order ..."

A19. Changed sentence accordingly (p.24, line 5).

**Corollary**

Author's comment: Additional to all the reviewers' suggestions, we decided to remove the paragraph about the circular shoal benchmark form the original manuscript in page 26, from line 11 to 17. With the introduction of several new benchmark problems (Reviewer #1 Question #7) and the modifications to the original manuscript, it felt unnecessary to keep this reference since the new results covered far more than what this benchmark offered.

[revised manuscript text omitted]

**Fig. 7 Mesh block computation using CUDA kernels. Each CUDA block is made of 64×4 threads and computes a portion of the mesh block. One CUDA thread computes one mesh block cell**

[Figure]

The kernel grid configuration for the SSWE is described briefly and shown in Fig. 7. CUDA threads are organized in two
10 dimensional blocks of size 64×4. The 64 threads in the *x* dimension cover the length of a mesh block requiring only one CUDA block. For the grid's *y* dimension, 16 CUDA blocks are set with 4 threads each, for a total of 16×4=64 threads, covering the height of the mesh block. With this configuration, one CUDA block computes a portion equal in size of the mesh block and the 16 CUDA blocks cover the entire mesh block. Additionally, one CUDA thread computes one mesh block cell. The specific calculation of each thread varies depending on the block type (*Wet*, *Dry*); however, the configuration
15 remains the same. In both cases, threads compute the governing equations described in section 3.1. The main difference occurs in the case of a *Dry* block; in this case, cells that represent land or coastline compute a reflective wall boundary.

[revised manuscript text omitted]
 purpose of applying this technique. Table 1 gathers the CFL numbers per level after implementing the sub-cycling. The second column shows the maximum $\Delta t$

10    allowed in each level using the initial CFL = 0.8. The third column shows the resulting number of sub cycles per level (ns) and the fourth column shows the new CFL values obtained for each level. In all cases the new CFL values remain below 1 to guarantee stability.

In general after a large $\Delta t$ step, corresponding boundary conditions are interpolated in time to update the sub steps. However this procedure introduces an additional computational overhead. To pursue the fastest modelling possible,

15    TRITON-G rescinds the boundary generation and instead uses the available boundary values at time *n*. Based on the benchmark and hindcast comparison, this decision proved to be acceptable based on the good agreement and accuracy of the results.

| Level | Max $\Delta t$ (CFL=0.8) | ns ($\Delta t$ =1.6) | S.C. CFL |
|-------|--------------------------|----------------------|----------|
| L1    | 10.71                    | 1                    | 0.12     |
| L2    | 5.13                     | 1                    | 0.25     |
| L3    | 2.37                     | 1                    | 0.54     |
| L4    | 1.65                     | 1                    | 0.78     |
| L5    | 0.95                     | 2                    | 0.68     |
| L6    | 0.55                     | 4                    | 0.59     |
| L7    | 0.26                     | 8                    | 0.39     |

**Table 1 CFL values used after introducing sub-cycling (S.C. CFL) for each of the seven levels. The second column**
20    **shows the maximum $\Delta t$ per level using CFL = 0.8 and the third column shows the number of sub-cycles (ns) required in each level when using $\Delta t$ =1.6**

Introducing this sub-cycling technique varies the GPU load initially created since a single block might be computed more than once. In order to guarantee load balance, two weights are applied to the space filling curve. The first weight takes

25    into account the different type of block and the second the number of sub-cycles. Each block gets attributed a *weight* during the SFC generation equal to the number of sub-cycles it requires. This approach for the domain partition allows to create a

fair work re-balance on the GPUs The effect of implementing the weighted load balance can be seen in Fig. 15 where GPU execution times per time-step are presented, with and without load balance. Implementation of the sub-cycling technique showed a speed up of around 15% in the total wall clock runtime.

[Figure]

**Fig. 15 GPU execution time with and without load balance**

**5.3.2 Runtime performance**

10    Several tests to estimate the performance of TRITON-G were done. Results ran on the Supercomputer *Tsubame* 3.0 (Tsubame, 2017) are presented, with Intel Xeon E5-2680 2.4GHz × 2, RAM 256GB, NVIDIA Tesla P100 (16GB) × 4/node, CUDA 8.0, gcc 4.8.5, Openmpi 2.1.1 and Omni-Path HFI 100 Gpbs network.

As comparison, results on a second machine are also presented, using four Tesla K80 (12GB×2) cards in a node (eight GPUs in total), GPUs are connected through PCI-Express 3.0, Intel Xeon CPU E5-2640 @2.6 GHz, RAM 128GB, CUDA

15    8.0, gcc 4.7.7 and Openmpi 1.8.6. These performance tests serve to show very good portability of our program on different hardware, older and much newer, without requiring changes or producing problems.

The breakdown of the main parts of the simulation using 3 GPUs is shown in Fig. 16, where *Inund* stands for Inundation kernel, *Wall* stands for the wall kernel, *Wet* for the *Wet* kernel and *X* and *Y* for the direction of the computation equivalent to longitude and latitude respectively. The process of updating the halos, presented in the graph as *Bnd* represent only 9% of the

20    total running time. It can be seen that the Wet and Wall kernel have similar performance despite the fact that the wall includes additional treatment for the coast boundaries. Since this treatment consists of many conditionals and they were replaced during optimization, it is understandable that the performance is similar. The slide *Others* include several values,

most importantly communications which represents around 1.5-2.0% of the total running time. ⬚formance of the main kernels on one GPU in floating point operations per second (FLOP/s) is gathered in Table 2.

| Kernel | GFLOP/s |
|--------|---------|
| WallX | 549.57 |
| WallY | 549.56 |
| WetX | 706.98 |
| WetY | 712.51 |
| Inund | 87.12 |

**Table 2. Kernel performance for one GPU in Giga FLOP per second**

Results for runtimes using Tesla P100 cards and Tesla K80 cards are presented in Fig. 17 for 1, 2, 3, 4 and 8 GPUs. For this test, 10 hours were simulated on the mesh initially generated for the Indian Ocean Domain (Fig. 5). All runtimes measurements include output time.

[Figure]

**Fig. 16 Computing breakdown shown in percentage**

A comparison between both GPU cards shows a speed up of almost 4 times from the older K80 cards to the latest P100 on Tsubame 3.0. In our collaboration project with RIMES an objective to complete this test under 15 minutes was set, which could be fulfilled by using 3 to 8 GPUs in this configuration. Run time for 3 GPU with K80 cards was 39.96 min and 12.1 min with P100 cards.

[Figure]

**Fig. 17 Wall clock comparison of 10-hour simulation on Tesla K80 and Tesla P100**

[Figure]

A saturation is noticeable in Fig. 17 as the number of GPUs are increased. A possible reason for this phenomenon is related to the increase of buffer preparation, packing/unpacking, and the communication exchange. Using the same domain size for all cases is another possible reason. Having fewer blocks on each GPU generates lower occupancy which might degrade performance. However, having met this study's time-to-solution objective of less than 15 minutes, no further optimization was deemed necessary.

By measuring the time required for the first wave to arrive in the focal areas, it was found that for Sri Lanka. Using 4 GPUs just 2 minutes wall clock time is required to generate the results of the inundation. The real tsunami wave took approximately 2 hours to propagate from the initial source to Sri Lanka, obtaining simulation results faster than real time gives authorities sufficient time to make decisions regarding evacuations.

**6 Tsunami inundation benchmark comparison**

In order to compare the numerical results of TRITON-G with existing benchmarks and test its ability to estimate inundation, we present the results obtained using the main benchmark tests proposed in the National Tsunami Hazard Mitigation workshop (NTHMP, 2012). Results from other models participating in the workshop can be consulted in that reference. In this section, the comparison of the benchmark "1993 Hokkaido-Nansei-Oki (Okushiri). Field" is shown. Further comparison results with benchmark problems 4, 6 and 7 (abbreviated as BP4, BP6, BP7) can be found in the appendix section.

A detailed description of the benchmarks can be found in (NTHMP, 2012) and the data needed for them can be found in the repository https://gitub.com/rjleveque/nthmp-benchmark-problems . For completeness we give a brief explanation of the benchmark and the tasks it involves.

[Figure]

**6.1 Benchmark Problem #9: Okushiri Island Tsunami - Field**

This benchmark problem (BP9) is based on the data collected from the Mw 7.8 Hokkaido-Nansei-Oki tsunami around Okushiri Island in Japan in 1993. The goal is to compare computed model results with the field measurements.

[Figure]

**Fig. 18 Left: entire domain refined mesh containing 7 levels. Right: zoom on Okushiri island. Higher resolution used around Monai Valley at level 7 (7 m approx.) and Aonae region at level 6 (14 m approx.)**

**6.1.1 Problem setup**

10  The following parameters were used for the computation:

- *Bathymetry*: taken from databases provided by (NTHMP, 2012), interpolated where necessary.
- *CFL*: 0.9
- *Simulated time*: 60 minutes
- *Initial condition*: source generated from the database provided by DCRC (Disaster Control Research Center) Japan
15  solution DCRC17a, described in (Takahashi, 1996).
- *Boundary conditions*: open boundaries at the four domain edges.
- *Friction*: Manning coefficient set to 0.02

- *Computational domain*: a mesh refinement is used on the entire domain (shown in Fig. 18). Seven levels are used in total. The resolution of base level 1 is 450 m and the resolution of level 7 is approximately 7 m. Dry blocks that did not take part in the computation were removed in the mesh generation process.

**6.1.2 Tasks to be performed**

This benchmark requires the following tasks to be performed:

8. Compute run-up around Aonae
9. Compute arrival of the first wave to Aonae
10. Show two waves at Aonae approximately 10 minutes apart; the first wave came from the wet, the second wave came from the east
11. Compute water level at Iwanai and Esashi tide gauges
12. Maximum modeled run-up distribution around Okushiri island
13. Modeled run-up height at Hamatsumae
14. Modeled run-up height at a valley north of Monai.

**6.1.3 Numerical results**

In this section we present the numerical results obtained with TRITON-G for benchmark problem #9.

**6.1.3.1 Run-up around Aonae**

The maximum inundation around Aonae peninsula modeled during the simulation is shown in Fig. 19. Contours every 4 meters are drawn to show the outline of the topography. Maximum inundation height computed was nearly 15 meters but the scale used is set to the upper limit of 10 m to highlight the areas where major inundation occurred.

[Figure]

**Fig. 19 Inundation map of Aonae region with 4-m contours of bathymetry and topography**

The west side of the peninsula received the impact of the first wave, which produced the largest inundation height. Maximum values of nearly 15 m were obtained in the simulation. Despite a relatively lower inundation height in the east side of the peninsula, deep penetration was found due to the flatter topography in this area. The inundation on the east side was mainly produced by the second wave coming from the east. The south side of the peninsula experienced the impact of both first and second waves and run-up of over 12 m was estimated.

**6.1.3.2 Arrival of first wave to Aonae**

The arrival of the first wave at Aonae peninsula is shown in Fig. 20. This wave is coming from the west. Snapshots are approximately 5 seconds apart at times 4.9 min and 5.0 min to illustrate the wave arrival. From these snapshots, we estimate that the wave made impact at around 5 minutes after the tsunami generation.

[Figure]

**Fig. 20 Arrival wave at Aonae peninsula coming from the west, snapshots of the wave at times 4.9 min and 5.0 min after tsunami generation**

5  **6.1.3.3 Two waves arriving at Aonae**

[Figure]

**Fig. 21 Two waves arriving at Aonae peninsula. Left: first wave coming from the west arrived at around t=5 min. Right: second wave coming from the east arrived at around t=16 min**

10  The two waves arriving at Aonae peninsula are shown in Fig. 21. The first one came from the west (Fig. 21 left) and made impact at around 5.0 min after the tsunami generation. The second major wave to hit the peninsula came from the east and made impact at around 16 min (Fig. 21 right). Slightly over 10 minutes separated the first and second wave.

**6.1.3.4 Tide gauge comparison at Iwanai and Esashi**

Comparison between computed and observed water levels at Iwanai and Esashi tide gauges is presented in Fig. 22. The arrival time of the computed wave shows good agreement for Esashi station. The computed wave positive and negative phases also follows rather well the observed values. In the case of Iwanai station the arrival time is slightly sooner than the observed however the observed wave phase is followed generally well in the computed results. The discrepancies between observed and computed values can be attributed to several reasons. Inaccuracies in the source used for the initial condition can influence greatly the result. Additionally, lack of realistic bathymetry including man-made structures around the area can affect the results as well.

[Figure]

**Fig. 22 Water level comparison between observations and TRITON-G results for Esashi (upper panel) and Iwanai (lower panel) tide gauges.**

Inserted in each panel of Fig. 22 are the estimated errors for the gauge comparison. The maximum wave amplitude error for Esashi station is 16.27% and for Iwanai 3.19%. These are considerably lower than the mean values obtained by the models reported in the workshop (NTHMP, 2012) of 43% and 36% respectively. Although no values are reported in (NTHMP, 2012), the NRMSD error is also estimated for our model and included in the panels, both values are under 20%.

**6.1.3.5 Maximum run-up around Okushiri**

The computed maximum run-up distribution around Okushiri Island is shown in Fig. 23. Observations were taken from (Kato and Tsuji, 1994). Good agreement is found between observed and computed values around the coast. Most values are within the observed range or within a small diference from the field measurement. The simulation seems to capture well the variations that occurred along the coast.

[Figure]

**Fig. 23 Computed and observed run-up values in meters along the coast of Okushiri island.**

The model could simulate well the maximum run-up observed around Monai valley within a reasonable 15% error. The major differences are found in the southwest side of the island where run-up values were underestimated with larger difference. The discrepancies could be explained by the use of different grid around the island coast. Additionally, the lack of an accurate high-resolution bathymetry database everywhere can also influence the computed values as well as an inaccurate initial condition.

**6.1.3.6 Run-up height at Hamatsumae**

The maximum inundation map for Hamatsumae region is shown in Fig. 24. Topography and bathymetry contours are outlined every 4 meters. A grid resolution of approximately 14 m was used for this region. Near the center of the region and to the east, run-ups of nearly 16 meters were computed. Additionally, inundation values ranging from 8 to 10 meters were obtained which match well with field observations.

[Figure]

**Fig. 24 Inundation map of Hamatsumae region with 4-m contours of bathymetry and topography.**

**6.1.3.7 Run-up height at a valley north of Monai**

The maximum inundation map for the valley north of Monai is shown in Fig. 25. Topography and bathymetry contours are outlined every 4 meters. A grid resolution of approximately 7 m was used for this region. Inundation of around 26 m was computed, relatively close to the 30.6 m observed in the field.

[Figure]

**Fig. 25 Inundation map for the valley north of Monai with 4-m contours of bathymetry and topography.**

[revised manuscript text omitted]

**Appendix**

Numerical results for benchmarks 4, 6 and 7 are presented in this section. Detail description of the problems can be found in

15   (NTHMP, 2012), we give a brief explanation in each section for completeness.

**A1 Benchmark problem #4: Solitary wave on a simple beach – Laboratory**

The domain for this test is shown in Fig. C. In this problem, the wave height H is located at a distance L from the beach toe. This test was replicated in a wave tank 31.73-cm-long, 60.96-cm-deep and 39.97-cm-wide at the California Institute of

20   Technology. Several experiments with different water heights were performed. Benchmark Problem 4 (BP4) uses the datasets for $H/d = 0.0185$ non-breaking wave and $H/d = 0.30$ breaking wave for code validation. Results use dimensionless units with the help of parameters like length $d$, velocity scale $U = \sqrt{gd}$ and time scale $T = \sqrt{d/g}$.

[Figure]

**Fig. A1 Domain sketch for BP4, slope 1:19.85 (figure taken from benchmark description)**

**A1.1 Problem setup**

- *Parameters*: d = 1, g = 9.8, case A with $H/d = 0.0185$ and case B with $H/d = 0.30$.

- *Friction*: Manning coefficient set to 0.01

- *Computational domain*: the domain along $x$ direction spanned from x $= -20$ to x $= 80$.

- *Boundary conditions*: a non-reflective boundary condition is used at the right side of the computational domain.

- *Grid resolution*: the numerical results presented are solved with a resolution of $\Delta x = 0.1$

- CFL: 0.9

- *Initial condition*: the initial wave is computed based on the following equations for height ($\eta$) and velocity (u) respectively

$$\eta(x.0) = Hsech^2[\gamma(x - x_s)/d],\qquad(22)$$

$$u(x,0) = -\eta(x,0)\sqrt{\frac{g}{d}}.\qquad(23)$$

**A1.2 Tasks to be performed**

To accomplish this problem, the following tasks should be performed:

1  Compare numerically calculated surface profiles at t/T=30:10:70 for the non-breaking case $H/d = 0.0185$ with the lab data (Case A).

2  Compare numerically calculated surface profiles at t/T=15:5:30 for the breaking case $H/d = 0.30$ with the lab data (Case C).

3  Compute maximum runups for at least one non-breaking and one breaking wave case.

**A1.3 Numerical results**

We present the numerical results obtained using TRITON-G. Figure A2 shows the comparison between water surface level measured in the experiment and the modeled numerical results obtained by our model for times 30, 40, 50, 60 and 70 for case A ($H/d = 0.0185$). Our results show good agreement between the numerical simulation and the non-breaking experiment.

Table A1 shows the errors computed for the normalized root mean square deviation (*NRMSD*) and for the maximum wave amplitude error (*MAX*). The error values obtained by the NTHMP workshop models are also included for comparison. These values are divided into two columns, one with results for the non-dispersive models (*ND*) and the other with results for the non-dispersive and dispersive models together (labeled *ALL*).

[Figure]

**Fig. A2. Comparison of numerically calculated free surface profile at different dimensionless times for the non-breaking case $H/d = 0.0185$.**

Errors obtained from our simulation tend to be similar or smaller than those errors obtained by other ND models, with just slight exception for time 70. Additionally, except for time 70 our errors are smaller than those obtained combining non-dispersive and dispersive mean error value.

Water level comparison for case C ($H/d = 0.30$) at times 15, 20, 25 and 30 is shown in Figure A2. Table A2 gathers the values for NRMSD and MAX errors for our numerical results and for the NTHMP workshop models. In this case, only the results of models that reported their errors are included (taken from Table 1-8, page 41 in (NTHMP, 2012) ).

For case C conditions, the shallow water equations are no longer appropriate for modeling and hydrostatic models tend to produce larger differences than non-hydrostatic ones. Our numerical results in general show good agreement with the experiment.

The difference with the steepening of the crest that is noticeable in the results is expected from a hydrostatic model. In spite of that, this steeping in our model is not very large and it can trace the wave front well. Once the wave breaking occurs, our model can simulate reasonably well the run-up. This is also partly reflected in the small NRMSD error estimation obtained by our model after the wave breaking.

Maximum run-up for case A and case C were calculated. For the non-breaking case A, the obtained run-up value is 0.091 and for the breaking case C the run-up estimated is 0.588. These values are plotted in Fig. A4 with a yellow and red dot respectively, it can be seen that both values lie well within the experimental results.

| | | NRMSD | | | MAX | | |
|---|---|---|---|---|---|---|---|
| | | TRITON-G | NTHMP | | TRITON-G | NTHMP | |
| $H = 0.0185$ | | | ND | ALL | | ND | ALL |
| T | 30 | 8.8 | 11 | 11 | 4.0 | 6 | 4 |
| T | 40 | 6.6 | 9 | 8 | 4.8 | 3 | 3 |
| T | 50 | 3.5 | 6 | 5 | 7.4 | 13 | 7 |
| T | 60 | 3 | 4 | 5 | 1.4 | 1 | 3 |
| T | 70 | 11 | 33 | 16 | 13.5 | 15 | 9 |

**Table A1. Model surface profile errors with respect to laboratory experimentso for case A $H/d = 0.0185$ at times 30, 40, 50, 60, and 70. Results from the NTHMP workshop errors are separated in non-dispersive (ND) models and all models (ALL).**

[Figure]

**Fig. A3.** Comparison of numerically calculated free surface profile at different dimensionless times for the breaking case $H/d = 0.30$.

| H = 0.3 | | NRMSD | | MAX | |
|---|---|---|---|---|---|
| | | TRITON-G | NTHMP ALL | TRITON-G | NTHMP ALL |
| T | 15 | 11.3 | 7 | 5.4 | 6 |
| T | 20 | 5.9 | 9 | 23.3 | 11 |
| T | 25 | 6.5 | 6 | 11.1 | 10 |
| T | 30 | 2.9 | 4 | 1.4 | 6 |

**Table A2. Modeled surface profile errors with respect to laboratory experimentso for case A $H/d\ =\ 0.30$ at times 15, 20, 25 and 30. Results from the NTHMP workshop model errors available are shown (ALL).**

[Figure]

**Fig. A4. Scatter plot of non-dimensional maximum run-up from a total of more than 40 experiments conducted by Y. Joseph Zhan** (Synolakis, 1987)**. Orange point indicates TRITON-G result for the breaking case $H/d\ =\ 0.30$ and yellow point indicates the result for the non-breaking run-up case $H/d\ =\ 0.0185$ .**

10   **A2 Benchmark problem #6: Solitary wave on a conical island – Laboratory**

The goal of this benchmark is to compare computed model results with laboratory measurements obtained during a physical modeling experiment conducted at the Coastal and Hydraulic Laboratory Engineer Research and Development Center of the U.S. Army Corps of Engineers. The laboratory physical model was constructed as an idealized representation of Babi Island, in the Flores Sea, Indonesia, to compare with Babi Island run-up measured shortly after the 12 December 15   1992 Flores Island tsunami (Yeh et al., 1994). Figure A5 show schematics of the experiment.

**A2.1 Tasks to be performed**

To accomplish this benchmark, it is suggested that, for

Case A: water depth d= 32.0 cm, target H=0.05, measured H=0.045

Case B: water depth d= 32.0 cm, target H=0.20, measured H=0.096

Case C: water depth d= 32.0 cm, target H=0.05, measured H=0.181

model simulations be conducted to address the following:

4. Demonstrate that two wave fronts split in front of the island and collide behind it
5. Compare computed water levels with laboratory data at gauge 6, 9, 16 and 22
6. Compare computed island run-up with laboratory gauge data

7.

**8. Fig. A5 Basin geometry and coordinate system. Solid lines represent approximate basin and wavemaker surfaces. Circles along walls and dashed lines represent wave absorbing material.**

**A2.2 Problem setup**

- *Computational domain*: [-5,23] × [0, 28]
- *Boundary condition*: open boundaries
- *Initial condition*: same solitary wave as proposed in BP4 with the correction for two dimensions.
- *Grid resolution*: the numerical results presented are solved with a resolution of $\Delta x = 0.05$

- *CFL*: 0.9
- *Friction*: Manning coefficient set to 0.02

[Figure]

*t = 31*

*t=32*

*t= 33*

*t=35*

**Fig. A6 Snapshots at several times showing the wavefront splitting in front of the island and colliding behind it for case B.**

**A2.3 Numerical results**

We present the numerical results obtained using TRITON-G for the three cases (A, B and C) except for the splitting-colliding item. For this item, Figure A6 shows the wave front splitting in front of the island and then colliding again behind it for case B (H=0.096), analogue behavior was obtained for the other two cases.

| | NRMSD | | | MAX | | |
|---|---|---|---|---|---|---|
| | TRITON-G | NTHMP | | TRITON-G | NTHMP | |
| Case A | | ND | ALL | | ND | ALL |
| Gauge 6 | 10 | 6 | 7 | 4 | 9 | 8 |
| Gauge 9 | 9 | 7 | 8 | 4 | 14 | 10 |
| Gauge 16 | 7 | 10 | 9 | 5 | 10 | 12 |
| Gauge 22 | 9 | 8 | 8 | 4 | 25 | 18 |
| Case B | | | | | | |
| Gauge 6 | 10 | 8 | 8 | 7 | 6 | 6 |
| Gauge 9 | 9 | 8 | 8 | 2 | 7 | 9 |
| Gauge 16 | 9 | 7 | 7 | 14 | 7 | 7 |
| Gauge 22 | 8 | 9 | 9 | 6 | 40 | 27 |
| Case C | | | | | | |
| Gauge 6 | 13 | 10 | 8 | 3 | 6 | 5 |
| Gauge 9 | 12 | 11 | 11 | 2 | 9 | 13 |
| Gauge 16 | 10 | 9 | 8 | 4 | 3 | 3 |
| Gauge 22 | 9 | 8 | 8 | 10 | 18 | 15 |

**Table A3. Water level time series TRITON-G model errors with respect to laboratory experiment data for case A, B and C. Mean values obtained for the performing NTHMP models is separated in non-dispersive models (ND) and non-dispersive and dispersive models together (ALL)**

Water level comparison uses values for gauges 6, 9, 16 and 22 for each of the 3 cases. Gauge 6 is located at $(9.36, 13.80, 31.7)$, Gauge 9 is located at $(10.36, 13.80, 8.2)$, Gauge 16 is located at $(12.96, 11.22, 7.9)$ and Gauge 22 is located at $(15.56, 13.80, 8.3)$.

[Figure]

5      **Fig. A7 Comparison between computed and measured water levels at gauges 6, 9, 16 and 22 for case A (H=0.045)**

[Figure]

5    Fig. A8 Comparison between computed and measured water levels at gauges 6, 9, 16 and 22 for case B (H=0.096)

[Figure]

5    **Fig. A9 Comparison between computed and measured water levels at gauges 6, 9, 16 and 22 for case C (H=0.181)**

[Figure]

5          **Fig. A10 Comparison between computed and measured run-up around the island for the three cases.**

Numerical results for Case A, B and C are shown in Fig. A7, Fig. A8 and Fig. A9 respectively. In the three cases results were stable and in good agreement with the experimental values. The incident wave height and arrival time was captured well for all gauges. Similarly as with BP4, the steepening of the wave with increasing H is expected in a non-hydrodynamic model.

5     After the wave hit the island, some differences between experimental and model wave are noticeable as the initial wave height increased. These oscillations in the experimental data represent the effects of dispersion, which our non-dispersive numerical method is not designed to capture. Despite this, the modeled waves show good agreement with the shape of the experimental waves and the errors estimated tend to be small.

| | | TRITON-G | NTHMP | | TRITON-G | NTHMP | |
|---|---|---|---|---|---|---|---|
| | | | ND | ALL | | ND | ALL |
| Runup | | NRMSD | NRMSD | | MAX | MAX | |
| Case | A | 9 | 18 | 18 | 0.6 | 12 | 7 |
| Case | B | 19 | 21 | 18 | 9 | 2 | 5 |
| Case | C | 20 | 12 | 11 | 14 | 5 | 5 |

**Table A4. Run-up TRITON-G model errors with respect to laboratory experiment data for case A, B and C. Mean values obtained for the performing NTHMP models is separated in non-dispersive models (ND) and all models (ALL) and presented for better comparison.**

    Table A3 gathers the normalize root mean square deviation (*NRMSD*) error and the maximum wave height (*MAX*) error. For comparison, mean errors obtained by the participating models in the NTHMP workshop are also included. These are separated in two columns, one for non-dispersive (ND) models and the other for non-disperse and disperse models together (ALL).

20     NRMSD errors for our model tend to be not very large and in similar range than those of the other non-dispersive models. In the case of the maximum height error (MAX), in almost all cases our model produced smaller error values than the non-dispersive model counterparts. Additionally, in most cases our MAX errors are smaller than those errors of the combined non-dispersive and dispersive mean values.

    Figure A10 shows the comparisons between computed and experimental run-up around the island for the three cases.

25 Case A represent the best agreement with the experimental values. Differences increased with steeper wave cases B and C as several reflections and refraction possibly occur in the basin.

    Table A4 gathers the errors obtained by our model and by the participating models in the NTHMP workshop for run-up cases A, B and C. Figure A10 showed the good agreement for Case A and this is also reflected in the NRMSD and MAX

error results. Both values are considerably smaller than those errors obtained by the NTHMP non-dispersive (ND) models and by the non-dispersive and dispersive together (ALL). For cases B and C, the errors tend to be larger but still similar to those obtained by other non-dispersive models. In all cases, the error stayed below the 20% recommended criteria.

5 **A3 Benchmark problem #7: The tsunami run-up onto a complex 3-D beach. Laboratory.**

A laboratory experiment using a large-scale tank at the central Research Institute for Electric Power Industry in Abiko, Japan was focused on modelling the runup of a long wave on a complex beach near the village of Monai (Liu et al., 2008). The beach in the tank was a 1:400-scale model of the bathymetry and topograpgy around a very narrow gully, where extreme runup was measured.

**A3.1 Problem setup**

The following parameters were used for the computation:

- *Grid resolution*: 393×244 was used with the same resolution 0.014 m as the bathymetry.
- *CFL*: 0.9
15 - *Initial condition*: water at rest.
- *Friction*: Manning coefficient set to 0.01
- *Boundary conditions*: Solid wall boundary were used at the top and bottom. At the left boundary, the given initial wave (shown in Fig. A11) was used to specify the condition up to time t=22.5 s, after that it became a wall boundary condition.

**Fig. A11 Prescribed input wave for the left boundary condition, defined from t=0 to t=22.5 s**

**A3.2 Tasks to be performed**

To accomplish this benchmark it is suggested to:

5. Model propagation of the incident and reflective wave accordingly to the benchmark-specified boundary condition.
6. Compare the numerical and laboratory-measured water level dynamics at gauges 5, 7 and 9.
7. Show snapshots of the numerically computed water level at the time synchronous with those of the video frames.
8. Compute maximum runup in the narrow valley.

**A3.3 Numerical results**

This section presents the numerical results for BP7 obtained with TRITON-G to achieve the required tasks.

The comparison with the three requested gauges 5, 7 and 9 is shown in Fig. A12 from $t = 0$ to $t = 25$ s. Good agreement is found between modeled and experimental wave for the three cases.

Values for the normalized Root Mean Square deviation error (NRMSD) and maximum wave amplitude error (MAX) were estimated for the gauge results. For gauge 5, the NRMSD error is 10% and MAX is 0.89%. For gauge 7, NRMSD is 10% and MAX is 4.81%. For gauge 9, the NRMSD error is 6.57% and MAX is 2.66%.

Comparison with the extracted movie frames is shown in Fig. A13. In the left column are the five frames provided from the laboratory recording. These are frames 10, 25, 40, 55 and 70, extracted from the video with a 0.5 s interval. We found good agreement in time and space for times 15 s to 17 s in 0.5 s increments, shown in the right column. The side-by-side comparison shows that the modeled wave follows the experimental wave front well. Additionally, the model captures the rapid run-up/run-down in the narrow gully.

Finally, the data provided by the benchmark workshop include a series of experiment tests for maximum run-up. Its maximum run-up is recorded at $\times = 5.1575$ and $y = 1.88$ m with an average value of approximately 0.09 m. In comparison, our numerical result recorded a maximum run-up at around $t = 16.5$ with a height of 0.0936 m at $\times = 5.15$ and $y = 1.88$ m.

[Figure]

**Fig. A12 Water level comparison for BP7 between experiment and TRITON-G for gauges 5, 7 and 9**

[Figure]

**Fig. A13 Comparison between extracted movie frames (left) and TRITON-G simulation (right) for times 15, 15.5, 16, 16.5 and 17 seconds**

**Data availability**

Underlying research data can be found in Open Science Framework repository:

[revised manuscript text omitted]

NTHMP: NTHMP, 2012. National Tsunami Hazard Mitigation Program (NTHMP). 2012. Proceedings and Results of the 2011 NTHMP Model Benchmarking Workshop, Department of Commerce/NOAA/NTHMP; NOAA Special Report., 2012.

20  NVIDIA: CUDA Zone, [online] Available from: https://developer.nvidia.com/cuda-zone, 2017a.

NVIDIA: Tesla P100 Datasheet, [online] Available from: https://images.nvidia.com/content/tesla/pdf/nvidia-tesla-p100-PCIe-datasheet.pdf, 2017b.

Nwogu, O.: An alternative form of the Boussinesq equations for nearshore wave propagation, Coastal, and Ocean Engineering, 119, 618–638, 1993.

25  Oceans (GEBCO), T. G. B. C. of the: GEBCO, [online] Available from: http://www.gebco.net/, 2017.

Ogata, Y. and Takashi, Y.: Multi-Dimensional Semi-Lagrangian Characteristic Approach to the Shallow Water Equations by the CIP Method, International Journal of Computational Engineering Science, 05(03), 2004.

Peregrine, D.: Long waves on a beach, Journal of Fluid Mechanics, 27(4), 815–827, 1967.

Plant, N., Kacey, E., Kaihatu, J., Veeramony, J., Hsu, L. and Todd, H.: The effect of bathymetric filtering on nearshore
30  process model results, Coastal Engineering, 56(4), 484–493, 2009.

Reed, D.: User Datagram Protocol (UDP), RFC 768., 1980.

Regional Integrated Multi-Hazard Early Warning System, R.: RIMES, [online] Available from: http://www.rimes.int/, 2017.

RIMES: Tsunami Hazard and Risk Assessment and Evacuation Planning - Hambantota, Sri Lanka, Regional Integrated Multi-Hazard Early Warning System., 2014.

Roeber, V. and Cheung, K. F.: Boussinesq-type model for energetic breaking waves in fringing reef enviroments, Coastal Engineering, 70, 1–20, 2012.

5  Rusanov, V.: Characteristics of the general equations of gas dynamics, Zhurnal Vychislistelnoi Mathematiki Mathematicheskoi Fiziki, 3, 508–527, 1963.

Sagan, H.: Space-Filling Curves, Universitext., 1994.

Shi, F., Kirby, J. T., Geiman, J. D. and Grilli, S.: A high-order adaptive time-stepping TVD solver for Boussinesq modeling of breaking waves and coastal inundation, Ocean Modeling, 43, 36–51, 2012.

10  Smylie, L. and Mansinha, D. E.: The Displacement Fields of Inclined Faults, B. Seismological Soc. Ame., 61(5), 1433–1440, 1971.

Srivihoka, P., Honda, K., Ruangrassamee, A., Muangsinc, V., Naparatb, P., Foytong, P., Promdumrong, N., Aphimaeteethomrong, P., Intaveec, A., Layug, J. E. and Kosinc, T.: Development of an online tool for tsunami inundation simulation and tsunami loss estimation, Continental Shelf Research, 79, 3–15, 2014.

15  Stoker, J. J.: Water Waves: The Mathematical Theory with Applications, Wiley-Interscience., 1992.

Supparsri, A., Koshimura, S. and Imamura, F.: Developing tsunami fragility curves based on the satellite remote sensing and the numerical modeling of the 2004 Indian Ocean tsunami in Thailand, J Natural Hazards and Earth Sc., 11, 173–189, 2011.

Swarztrauber, P. N., Williamson, D. L. and Drake, J. B.: The cartesian method for solving partial differential equations in spherical, Dynamics of Atmospheres and Oceans, 27, 679–706, 1997.

20  Synolakis, C. E.: The runup of solitary Waves, J. Fluid Mechanics, 185, 523–545, 1987.

Szauer, G.: Game Physics Cookbook, Amazon Digital Services., 2017.

Takahashi, T.: Benchmark Problem 4. The 1993 Okushiri tsunami. Data collected, Conditions and Phenomena, in Long waves runup models, edited by H. Yeh, P. Piu, and C. Synolakis, pp. 384–403, Word Scientific Publishing Co., 1996.

Thacker, W. C.: Some exact solutions to the nonlinear shallow-water wave equations, J. Fluid Mechanics, 107, 499–508, 25  1981.

Titov, V. and Synolakis, C.: Evolution and runup of the breaking and nonbreaking waves using VTSC2, J. Waterway, Port, Coastal and Ocean Eng., 126(6), 308–316, 1995.

Titov, V., Rabinovich, A., Mojfeld, H., Thomson, R. and Gonzales, F.: The Global Reach of the 26 December 2004 Sumatra Tsunami, Science, 309(5743), 2045–2048, 2005.

30  Toro, F.: Shock-capturing methods for free-surface shallow flows, John Wisley&Sons AK Peters AK Peters Ltd., 2010.

Tsubame, T. I. of T.: Manual Tsubame 3.0, [online] Available from: http://www.t3.gsic.titech.ac.jp/, 2017.

Utsumi, T., Kunugi, T. and Aoki, T.: Stability and accuracy of the cubic interpolated propagation scheme, Computer Physics Communications, 101, 9–20, 1997.

Vazhenin, A., Lavrentiev, M., Romanenko, A. and Marchuk, A.: Acceleration of tsunami wave propagation modeling based on re-engineering of computational components, International Journal of Computer Science and Network Security, 13, 32– 70, 2013.

Vincent, S., Caltagirone, J. P. and Bonneton, P.: Numerical modeling of bore propagation and run-up on sloping beaches using a MacCormack TVD scheme, Journal of Hydraulic Research, 39, 41–49, 2001.

Wang, D., Becker, N. C., Walsh, D., Fryer, G. J., Weinstein, S. A., McCreery, C. S., Sardina, V., Hsu, V., Hirshorn, B. F., Hayes, G. P., Duputel, Z., Rivera, L., Kanamori, H., Koyangai, K. and Shiro, B.: Real-time forecasting of the April 11, 2012, Sumatra Tsunami, Geophys. Res. Lett., 39(19), L19601, 2012.

Wei, G., Kirby, J., Grilli, S. T. and Subramanya, R.: Fully nonlinear Boussinesq model for free surface waves. Part 1: Highly nonlinear unsteady waves, J Fluid Mech, 294, 71–92, 1995.

WHO, W. H. O.: Indonesia situation reports, [online] Available from: http://www.who.int/hac/crises/idn/sitreps/en/, 2014.

Williamson, D. L., Drake, J. B., Hack, J. J., Jakob, R. and Swarztraube, P. N.: A standard test set for numerical approximations to the shallow water equations in spherical geometry, J. Comput. Phys, 102, 211–224, 1992.

Yabe, T. and Aoki, T.: A universal solver for hyperbolic equations by Cubic-Polynomial Interpolation I. One-dimensional solver, Comp. Physic Comm, 66, 219–232, 1991.

Yabe, T., Tanaka, R., Nakamura, T. and Xiao, F.: An Exactly Conservative Semi-Lagrangian Scheme (CIP–CSL) in One Dimension, Monthly Weather Rev., 129, 332–344, 2001.

Yamamoto, S. and Daiguji, H.: Higher-order-accurate upwind schemes for solving the compressible Euler and Navier-Stokes equations, Computers and Fluids, 22(2), 259–270, 1993.

Yamazaki, Y., Cheung, K. F. and Kowalik, Z.: Depth-integrated, non-hydrostatic model with grid nesting for tsunami generation, propagation, and run-up, International Journal for Numerical Methods in Fluids, 67(12), 2081–2107, 2011.

Yeh, H., Liu, P., Briggs, M. and Synolakis, C.: Propagation and amplification of tsunamis at coastal boundaries, Nature, 372, 353–355, 1994.

Yerry, M. and Shephard, M.: Automatic three-dimensional mesh generation by the modified-octree technique, J. Numer. Meth. Eng., 32(4), 709–749, 1991.

Zaytsev, A., Yalciner, A., Chernov, A., Pelinovsky, E. and Kurkin, A.: NAMI DANCE, [online] Available from: http://namidance.ce.metu.edu.tr, 2006.

Zhang, Y. and Baptista, A. M.: An efficient and robust tsunami model on unstructured grids, Pure and Applied Geophysics, 165, 2229–2248, 2008.

Zhou, J. G., Causon, M. D., Mingham, C. and Ingram, G.: The surface gradient method for the treatment of source terms in the shallow-water equations, J Comp. Physics, 168, 1–52, 2001.